# ELAVL1a is an immunocompetent protein that protects zebrafish embryos from bacterial infection

Shousheng Ni[1], Yang Zhou[1], Lili Song[1], Yan Chen[1], Xia Wang[1], Xiaoyuan Du[1] & Shicui Zhang [1,2✉]

Previous studies have shown that ELAVL1 plays multiple roles, but its overall biological function remains ill-defined. Here we clearly demonstrated that zebrafish ELAVL1a was a lipoteichoic acid (LTA)- and LPS-binding protein abundantly stored in the eggs/embryos of zebrafish. ELAVL1a acted not only as a pattern recognition receptor, capable of identifying LTA and LPS, as well as bacteria, but also as an effector molecule, capable of inhibiting the growth of Gram-positive and -negative bacteria. Furthermore, we reveal that the C-terminal 62 residues of ELAVL1a positioned at 181–242 were indispensable for ELAVL1a antibacterial activity. Additionally, site-directed mutagenesis revealed that the hydrophobic residues Val192/Ile193, as well as the positively charged residues Arg203/Arg204, were the functional determinants contributing to the antimicrobial activity of rELAVL1a. Importantly, microinjection of rELAVL1a into embryos markedly promoted their resistance against pathogenic *Aeromonas hydrophila* challenge, and this pathogen-resistant activity was considerably reduced by co-injection of anti-ELAVL1a antibody or by knockdown with morpholino for *elavl1a*. Collectively, our results indicate that ELAVL1a is a maternal immune factor that can protect zebrafish embryos from bacterial infection. This work also provides another angle for understanding the biological roles of ELAVL1a.

[1] Institute of Evolution & Marine Biodiversity and Department of Marine Biology, Ocean University of China, Qingdao 266003, China. [2] Laboratory for Marine Biology and Biotechnology, Pilot National Laboratory for Marine Science and Technology (Qingdao), Qingdao 266003, China. ✉email: sczhang@ouc.edu.cn

ELAVL/Hu is a family of RNA-binding proteins (RBPs) that share high homology to the *Drosophila* ELAV (embryonic lethal-abnormal vision) gene[1,2]. In mammals, the ELAVL/Hu family consists of four members, i.e., HuR or HuA, HuB or Hel-N1, HuC and HuD[3–5] that correspond to non-mammalian ELAVL1, ELAVL2, ELAVL3, and ELAVL4, respectively[6–10]. All ELAV/Hu family members contain RNA-recognition motifs (RRMs) for specific uridine (U)-rich RNA structures. ELAVL2/HuB/Hel-N1, ELAVL3/HuC, and ELAVL4/HuD proteins are mostly distributed in neurons, while ELAVL1/HuR/HuA protein widely distributed in both neuronal and non-neuronal tissues[11,12].

ELAVL1/HuR/HuA is a well-characterized member of the ELAV/Hu family, and contains three RRMs that are central components of their RNA-binding domains[13]. HuR/HuA regulates gene expression at post-transcriptional level by stabilizing target mRNA[14]. Mammalian HuR/HuA has been shown necessary for proper embryonic development and stem/progenitor cell survival in mice[15,16], and associated with various tumor biological characteristics, including tumor development and progression, invasion, migration, prognosis, and chemotherapy resistance[17–19]. Accumulating data also demonstrate that HuR/HuA is been associated with inflammatory/immune response in mice[20–22]. Furthermore, HuR/HuA is found to interact directly with mRNAs encoding immune factors such as CD3 and TNF[23–27]. All these suggest that ELAV1/HuR/HuA plays an important role in immune response, but its exact function in immunity remains poorly defined. Moreover, only a limited number of studies on the function of ELAV1/HuR/HuA has been largely restricted to mammalian species, and information as such is rather scarce in other animal models.

Interestingly, two ELAV1-like genes, *elavl1a* and *elavl1b*, are identified from zebrafish *Danio rerio*[28]. It is found that *elavl1a*, which is more similar to HuR/HuA gene, represents the major expressed isoform during embryogenesis, and is required for erythropoiesis[28,29]. We have recently isolated a protein from zebrafish embryos by lipoteichoic acid (LTA)-conjugated Sepharose CL-4B affinity chromatography, and identified it as ELAVL1a[30]. Given that LTA is a signature molecule of Gram-positive bacteria, this obviously suggests an interaction between ELAVL1a and bacteria. However, we know nothing about it at present. Therefore, the aims of this study are to explore if ELAVL1a exhibits any antibacterial activity, and if so, to examine its mode of action, using zebrafish as a model.

## Results

### ELAVL1a is an LTA-binding protein stored in early embryos.
As reported previously by Ni et al.[30], the proteins eluted from LTA-conjugated Sepharose CL-4B column were resolved by SDS-PAGE, and six main bands (1 to 6) identified. MALDI/TOF MS analysis revealed that the band 5 was ELAVL1a (Fig. 1a), consisting of 91 amino acids, with the sequence coverage of 7%. In parallel, we showed that the mouse anti-human ELAVL1 antibody (ABIN577055) was not only reactive with the recombinant zebrafish ELAVL1a protein but also reactive with the natural ELAVL1a protein from zebrafish tissues (skin and muscle) and embryos as well as the natural ELAVL1a protein from mouse muscle (Supplementary Fig. 1). On one hand, these results confirmed that the protein in the band 5 was indeed ELAVL1a. On the other hand, they qualitied the antibody for the following experiments, as it reacted with both human and mouse as well as zebrafish ELAVL1a, as claimed by the product description. Western blotting showed that in zebrafish, ELAVL1a was present in the heart, liver, muscle, skin, brain, gill, ovary, and testis (Fig. 1d), with a relatively higher levels in the heart, ovary, and

testis. ELAVL1a was also detected in the fertilized eggs, embryos, and 7-day-old larvae, though its content decreased with development (Fig. 1e). These indicated that ELAVL1a was an LTA-binding protein distributed widely in different tissues and stored abundantly in early embryos of zebrafish.

### ELAVL1a is ubiquitously distributed in embryonic cells.
Immunohistochemical examination showed that ELAVL1a was localized in the blastoderm of 2-, 8-, 16-, and 256-cell embryos (Fig. 2a), but little positive signal (green fluorescence) was seen in the yolk sac. Moreover, no positive signal was seen in the control embryos immunostained with the isotype antibody instead of the primary antibody (Supplementary Fig. 2). These suggested that ELAVL1a was widely distributed in embryonic cells.

### Structure and phylogenetics of *elavl1a*.
ELAVL1a cDNA we obtained was 2160 bp long and contained an open reading frame (ORF) of 975 bp, a 5′-untranslated region (UTR) of 58 bp, and a 3-UTR of 1127 bp (Supplementary Fig. 3). The ORF coded for a protein of 324 amino acids with a calculated MW of ~35.92 kDa and a pI of 8.99. ELAVL1a included three RRMs positioned at residues 19–92, 105–180, and 243–316, respectively (Fig. 1b). ELAVL1a had no typical nuclear localization signals (Supplementary Table 1). Sequence alignment showed that ELAVL1a was 85.5–88.8%, 85.5–85.8%, 86.1%, 83.0–83.6%, 85.7–88.0%, 65.3%, and 38.6% identical to mammalian, avian, reptile, amphibian, fish, amphioxus and coral counterparts (Supplementary Fig. 4), and 85.7%, 69.8%, 69.8%, and 70.4% identical to ELAVL1b, ELAVL2, ELAVL3, and ELAVL4 (Supplementary Fig. 5). Structure prediction using SWISS-MODEL online software revealed that the 3D structure of ELAVL1a was similar to that of human HuR/HuA (ELAV1), both consisting of 5 α-helices and 11 β-sheets (Fig. 1c). The phylogenetic tree constructed using the available sequences of ELAV1 proteins showed that ELAVL1a and ELAVL1b formed an independent clade, which grouped together with other teleost ELAVL1 proteins and located in between tetrapod and invertebrate counterparts (Supplementary Fig. 6), well reflecting the established phylogeny of chosen organisms.

### Tissue- and stage-specific expression of *elavl1a*.
In adult zebrafish, *elavl1a* was expressed primarily in the eye, liver, ovary, and brain in a tissue-specific manner, with the highest expression in the ovary (Fig. 2b). In embryos/larvae, *elavl1a* mRNA was abundant in the zygotes, which decreased continuously with development, but remained detectable in 7-day-old larvae (Fig. 2c). Figure 2d shows the expression pattern of *elavl1a* in developing embryos/larvae. Pan-expression of *elavl1a* was observed in the embryos before segmentation stages, and then *elavl1a* mRNA was mainly restricted to the head region and muscle of 1-, 2-, and 3-day-old larvae (Fig. 2d) with the brain, eyes, and muscle as the main organs at these stages. These suggested that ELAVL1a might play a role in the formation of the head including the brain and eyes and muscle.

### rELAVL1a has affinity to bacteria, LTA, LPS, and lipid A.
ELAVL1a was expressed in *E. coli* BL21 (DE3) and purified by affinity chromatography on a Ni-nitrilotriacetic acid resin column. The purified recombinant protein, rELAVL1a, yielded a single band of ~40.0 kDa on SDS-PAGE after Coomassie Blue staining, corresponding to the expected size (Supplementary Fig. 7). Western blotting showed that the purified rELAVL1a reacted with the mouse anti-His-tag antibody, indicating that rELAVL1a was properly expressed.

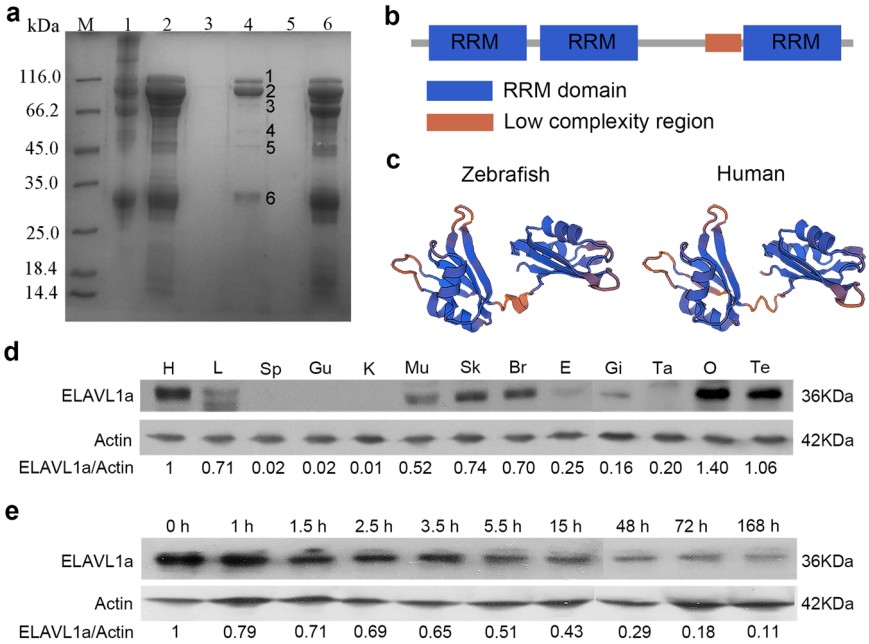

**Fig. 1 Identification of ELAVL1a as an LTA-binding protein. a** SDS-PAGE of the proteins isolated from the embryos extracts of zebrafish on LTA-conjugated Sepharose CL-4B affinity resin. Lane M, marker; lane 1, embryos extracts; lane 2, effluent fractions after Tris-HCl wash; lane 3, effluent fractions after throughout elution; lane 4, effluent fractions containing the absorbed proteins; lane 5, effluent fractions from LTA-conjugated Sepharose CL-4B affinity resin with no embryos extracts loaded; lane 6, effluent fractions from Sepharose CL-4B affinity resin with no conjugated LTA. **b** Domain structure of ELAVL1a predicted by the SMART program. **c** 3D structures of the ELAVL1a generated by SWISS-MODEL online software using human ELAVL1 (PDB code: 4egl.1.A) as the model. **d** Western blotting of ELAVL1a in different tissues including heart (H), liver (L), spleen (Sp), gut (Gu), kidney (K), muscle (Mu), skin (Sk), brain (Br), eye (E), gill (Gi), ovary (O), testis (Te) and tail (Ta). **e** Western blotting of ELAVL1a at the different developmental stages including zygote (0 h), 4-cell stage (about 1 h), 16-cell stage (~1.5 h), 256-cell stage (~2.5 h), high blastula stage (~3.5 h), 50% epiboly stage (~5.5 h), 10-somite stage (~15 h), 2-day post-fertilization (2-dpf), 3-dpf, and 7-dpf. The values of the quantification of Western blotting were given as well. Data are presented as the means ± SD.

As shown in Fig. 3a, b, rELAVL1a had strong affinity to the Gram-positive bacteria *M. luteus*, *B. subtilis*, and *S. aureus* and the Gram-negative bacteria *E. coli*, *V. anguillarum* and *A. hydrophila*. By contrast, TRX-His-tag peptide used as control did not show affinity to the same bacteria (Supplementary Fig. 8). In agreement with the above observations, FITC-labeled rELAVL1a bound to all the bacteria tested, but the FITC-labeled TRX-His-tag peptide did not (Supplementary Fig. 9). These indicated that rELAVL1a was able to interact with both the Gram-positive and -negative bacteria. This not only supported that ELAVL1a was an LTA-binding protein but also suggested it was an LPS-binding protein.

We then tested if rELAVL1a could bind to LTA and LPS by ELISA. As shown in Fig. 3c, d, rELAVL1a bound to both LTA and LPS in a dose-dependent manner, whereas BSA or TRX-His peptide used as control did not. This was further corroborated by the pull-down experiments, which showed that rELAVL1a bound to both LTA- and LPS-conjugated resin column, but recombinant TRX-His tag did not (Supplementary Fig. 10). Furthermore, the competition assay showed that the interaction of rELAVL1a with LPS was clearly inhibited by LTA or LPS in a dose-dependent manner (Fig. 3g, h). These showed without a doubt that ELAVL1a was indeed an LTA- and LPS-binding protein. In addition, we also demonstrated that the FITC-labeled rELAVL1a which was pre-incubated with LTA or LPS displayed reduced binding to the Gram-positive and -negative bacteria, respectively (Supplementary Fig. 9). These data suggested that ELAVL1a was able to specifically interact with the Gram-positive and -negative bacteria via LTA and LPS, respectively.

ELISA also revealed that rELAVL1a had a strong affinity to lipid A (Fig. 3e), a core component of LPS. Besides, the binding of rELAVL1a to LPS was not inhibited by the sugars examined, even at a concentration as high as 20 mg/ml (Fig. 3f). These indicated that rELAVL1a interacted with LPS mainly via lipid A, and had little lectin activity.

**rELAVL1a is an antibacterial protein**. As zebrafish ELAVL1a was an LTA- and LPS-binding protein, we thus wondered if it had any antibacterial activity. It was found that rELAVL1a not only inhibited the growth of all the Gram-positive and -negative bacteria tested, including the intracellular bacterium *Edwardsiella tarda*, in a dose-dependent manner (Fig. 4a and Supplementary Fig. 11a) but also caused different sizes of halo in the agar plate containing the bacteria (Fig. 4b and Supplementary Fig. 11b). These indicated that in addition to binding to LTA and LPS, rELAVL1a had a wide spectrum of antibacterial activity. At 50 μg/ml, the antibacterial activity of rELAVL1a against the Gram-positive and -negative bacteria was nearly comparable to that of 2 μg/ml kana. Transmission electron microscopy (TEM) examination revealed that rELAVL1a induced direct damage to the bacterial cells (Fig. 4c), i.e., the cells were swollen, the cell surfaces dissolved, the membranes disrupted and the cytoplasm became less dense. These indicated that ELAVL1a is an antibacterial agent capable of directly inhibiting growth of Gram-positive and -negative bacteria.

**rELAVL1a causes membrane depolarization and permeabilization**. As rELAVL1a had antibacterial activity, we thus tentatively tested its mode of action. Compared with control, the fluorescence intensities of all the bacterial cells (*M. luteus*, *B. subtilis*, *S. aureus*, *E. coli*, *V. anguillarum* and *A. hydrophila*)

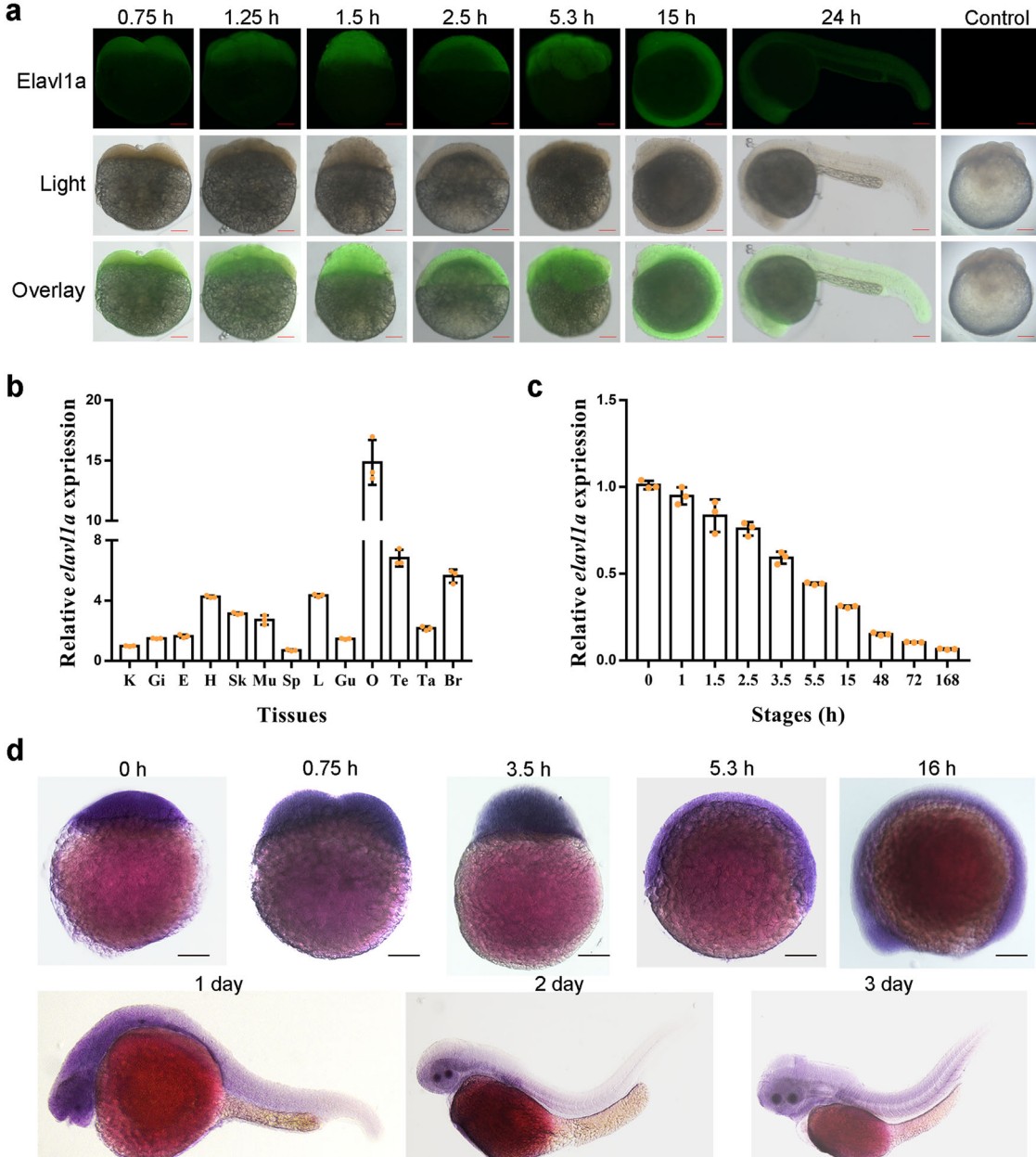

**Fig. 2 Distribution of zebrafish ELAVL1a in the cells of different developmental stage embryos and expression patterns of zebrafish *elavl1a* in the different tissues as well as at the different developmental stages. a** Immunohistochemical localization of ELAVL1a in the different developmental stage: 2-cell stage (about 0.75 h); 8-cell stage (~1.25 h); 16-cell stage (~1.5 h); 256-cell stage (~2.5 h); 50% epiboly stage (~5.3 h); 10-somite stage (~15 h); 24-h post-fertilization (24-hpf); Control: 16-cell embryo incubated with mouse pre-immune serum as control. Scale bars represent 100 μm. **b** Expression profiles of zebrafish *elavl1a* in the different tissues including kidney (K), gill (Gi), eye (E), heart (H), skin (Sk), muscle (Mu), spleen (Sp), liver (L), gut (Gu), ovary (O), testis (Te), tail (Ta), and brain (Br). **c** Expression profiles of zebrafish *elavl1a* at the different developmental stages including zygote (0 h), 4-cell stage (about 1 h), 16-cell stage (~1.5 h), 256-cell stage (~2.5 h), high blastula stage (~3.5 h), 50% epiboly stage (~5.5 h) embryos, 10-somite stage (~15 h), 2-day post-fertilization (2-dpf), 3-dpf, and 7-dpf larvae. *β-actin* was chosen as the internal control for normalization. Relative expression data were calculated by the method of $2^{-\Delta\Delta Ct}$. The vertical bars represent the mean ± SD ($n = 3$). The data are from three independent experiments performed in triplicate. **d** Expression of *elavl1a* during early development detected by WISH. Stages of embryonic development: newly fertilized egg (0 h); 2-cell stage embryo (~0.75 h); high blastula stage embryo (~3.5 h); 50% epiboly stage embryo (~5.3 h); 14-somite larvae (16 h); 1-day-old larvae; 2-day-old larvae; 3-day-old larvae. The color of the positive signal is purple.

treated with rELAVL1a increased (Fig. 5a and Supplementary Fig. 12a). The increase in fluorescence intensities was induced by the DiSC$_3$–5 probe released to the medium from intact cytoplasmic membrane, which thus indicated that rELAVL1a caused depolarization of the bacterial plasma membrane[31]. In addition, flow cytometry revealed that few of the non-treated bacterial cells

(*M. luteus*, *B. subtilis*, *S. aureus*, *E. coli*, *V. anguillarum* and *A. hydrophila*) had a PI fluorescent signal (Fig. 5b, Supplementary Fig. 12b and Supplementary Fig. 13), indicating that they had intact and viable cell membranes. By contrast, a significant proportion of the cells treated with rELAVL1a displayed a fluorescent signal, and the number of cells with a fluorescent signal increased

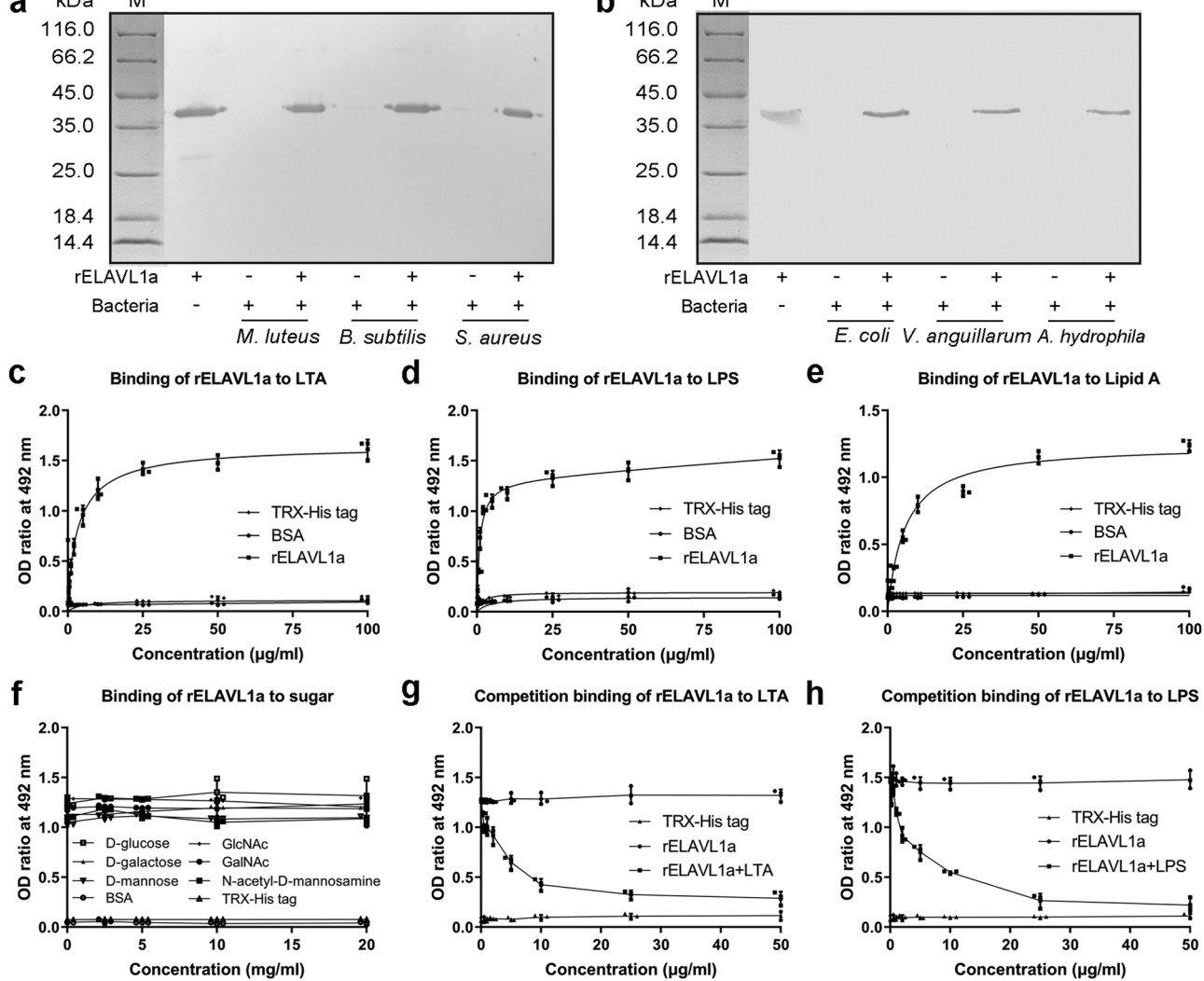

**Fig. 3 Binding of rELAVL1a to Gram-positive and Gram-negative bacteria, LTA, LPS, and lipid A. a, b** Western blotting about interaction of rELAVL1a with Gram-positive (**a**) and Gram-negative (**b**) bacteria. Lane M, marker; lane 1, purified rELAVL1a; lane 2, 4 and 6, *M. luteus*, *B. subtilis* and *S. aureus* (**a**) or *E. coli*, *V. anguillarum* and *A. hydrophila* (**b**) incubated in the presence of rELAVL1a; lane 3, 5, and 7, *M. luteus*, *B. subtilis* and *S. aureus* (**a**) or *E. coli*, *V. anguillarum* and *A. hydrophila* (**b**) incubated in the absence of rELAVL1a. **c, d, e** Interaction of rELAVL1a with LTA (**c**), LPS (**d**), Lipid A (**e**) revealed by ELISA. **f** Effects of various sugars on the interaction of rELAVL1a with LPS. **g, h** Interaction of rELAVL1a with LPS was inhibited by LTA (**g**) and LPS (**h**), the final concentration of rELAVL1a was 25 μg/ml. Each point in the graph represents the mean ± S.D. (*n* = 3). The data are from three independent experiments performed in triplicate. The bars represent the mean ± S.D.

with increasing doses of rELAVL1a, indicating that their cell membranes were no longer intact and became permeable to PI. These data suggested that rELAVL1a disrupted the bacterial membranes by a combined action of membrane depolarization and permeabilization.

**ELAVL1a plays role in antibacterial defense of early embryos.** The protein concentration of embryonic extracts prepared from 64- and 128-cell embryos was approximately 5 mg/ml. The embryonic extracts showed conspicuous antibacterial activity against the Gram-positive bacteria *M. luteus*, *B. subtilis,* and *S. aureus* and the Gram-negative bacteria *E. coli*, *V. anguillarum* and *A. hydrophila* (Fig. 6 and Supplementary Fig. 14). Interestingly, pre-incubation of the extracts with mouse anti-ELAVL1a antibody (ELAVL1aAb) significantly reduced this bacterial growth-inhibitory activity, but pre-incubation with anti-β-actin antibody (AcAb) did not (Fig. 6a and Supplementary Fig. 14). These findings suggested that ELAVL1a was a molecule

directly associated with the antibacterial activity of the embryo extracts.

To examine whether ELAVL1a could play a similar role in vivo, 8-cell embryos were each microinjected with ELAVL1aAb to block ELAVL1a action, followed by injection with live *A. hydrophila*. As shown in Fig. 6b, the majority (~90%) of the embryos injected with PBS, BSA, AcAb, ELAVL1aAb, rELAVL1a, rELAVL1a plus AcAb or rELAVL1a plus ELAVL1aAb developed normally, with the 24-h cumulative mortality rate of about 10%. Notably, the challenge with live *A. hydrophila* resulted in an increase in the mortality rate of the embryos injected with ELAVL1aAb (resulting in a decrease of ELAVL1a), with a 24-h mortality rate of ~85.2%, whereas the same challenge caused only ~68.9%, 69.2%, and 69.3% mortality at 24-h in the embryos injected with either PBS, BSA or AcAb alone. This indicated that blocking of ELAVL1a action in the embryos was able to reduce their anti-*A. hydrophila* activity. In comparison, the 24-h mortality rate of the embryos injected with rELAVL1a (resulting in an increase of ELAVL1a) was reduced to ~54.4%, which was

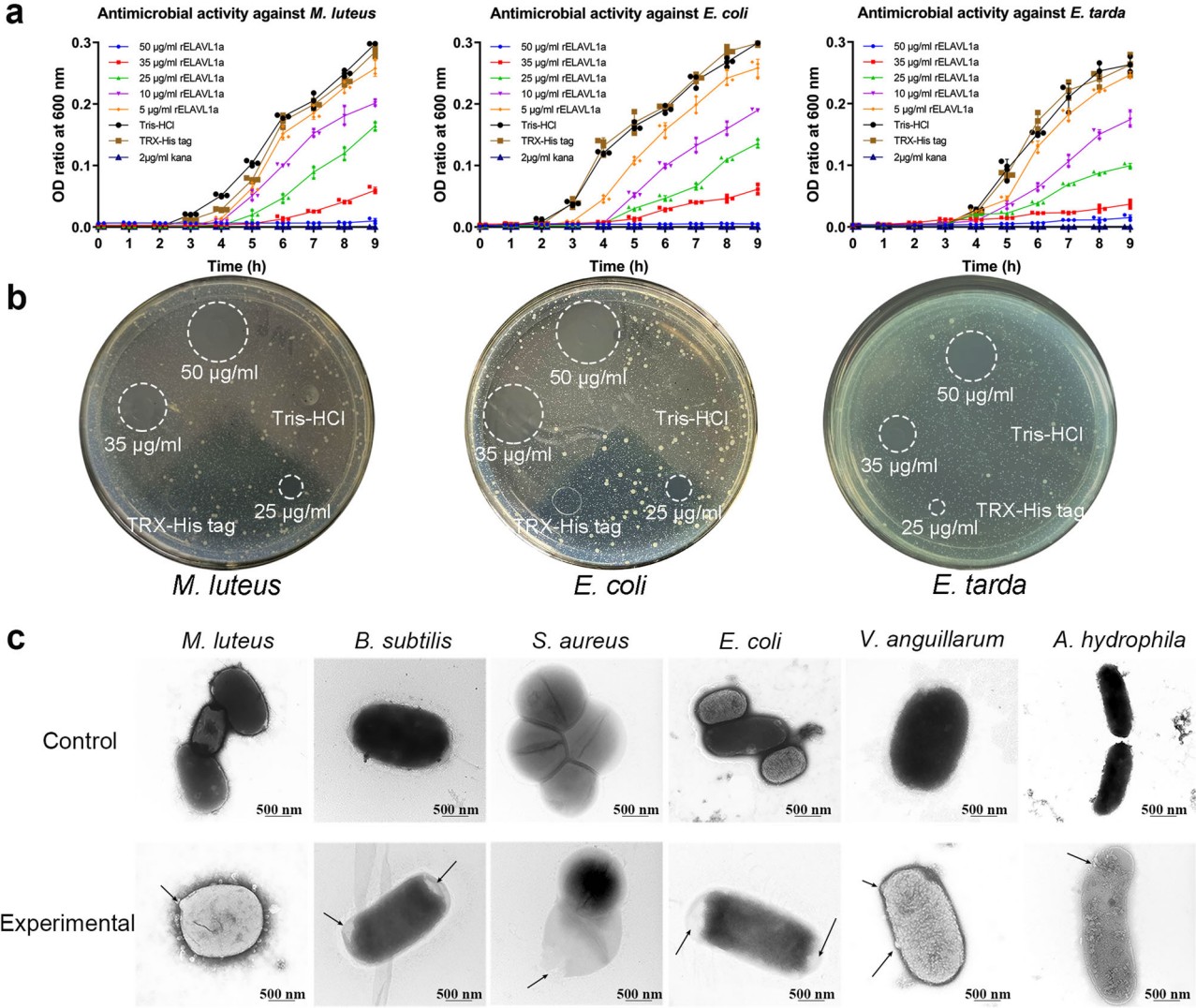

**Fig. 4 Antibacterial activity of rELAVL1a and its effects on bacterial structures. a** Antibacterial activities of rELAVL1a against Gram-positive bacteria *M. luteus* and Gram-negative bacteria *E. coli* and *E. tarda*. Each point in the graph represents the mean ± S.D. (*n* = 3). The data are from three independent experiments performed in triplicate. The bars represent the mean ± SD. **b** The halos showing antibacterial activities of rELAVL1a against representative bacteria, including the Gram-positive bacterium *M. luteus* and the Gram-negative bacteria *E. coli* and *E. tarda*. The halos without the cup (bottom) were shown. **c** Transmission electron microscopy. Control, *M. luteus*, *B. subtilis*, *S. aureus*, *E. coli*, *V. anguillarum* and *A. hydrophila* incubated with PBS; Experimental, *M. luteus*, *B. subtilis*, *S. aureus*, *E. coli*, *V. anguillarum* and *A. hydrophila* incubated with rELAVL1a. Scale bars represent 500 nm.

markedly lower than that of the embryos injected with AcAb or BSA or PBS alone, suggesting that the increased amount of ELAVL1a contributed to the protection of embryos against *A. hydrophila* attack. Moreover, the protecting activity of rELAVL1a against *A. hydrophila* in the embryos was apparently reduced by co-injection with ELAVL1aAb, but not by co-injection with AcAb (24-h mortality rate being ~84.8% versus ~56.7%), suggesting the specificity of ELAVL1a antibacterial activity. Moreover, we showed that the synthesis of ELAVL1a was reduced in the embryos microinjected with *elavl1a*-MO, indicating a successful knockdown of ELAVL1a (Fig. 6c). As shown in Fig. 6d, the 24-h mortality of *elavl1a*-MO-knockdown embryos increased up to 79%, in contrast to 63% mortality of control embryos, after challenge with live *A. hydrophila*, and this increase in mortality could be rescued by co-injection of *elavl1a*-MO with *elavl1a*-mRNA. This provided additional evidence that ELAVL1a played a protective role in the early embryos.

We then wondered if the embryos contained a sufficient endogenous amount of ELAVL1a to inhibit bacterial growth in vivo. ELISA revealed that ELAVL1a contents measured in a single one of the newly fertilized eggs, 12- or 24-h embryos were about 66.2 µg/ml, 58.3 µg/ml, and 57.7 µg/ml, individually. It was clear that ELAVL1a level in each egg or embryo was all markedly higher than 50 µg/ml rELAVL1a, a concentration necessary to inhibit the growth of Gram-positive and -negative bacteria in vitro. Thus, the endogenous concentration of ELAVL1a in each egg or embryo was sufficient enough to kill potential pathogens in vivo, at least at the initial 24 hpf. All data above suggested that ELAVL1a was an antibacterial protein capable of protecting zebrafish early embryos from bacterial infection.

**C-terminal 62 residues at 181–242 are critical for antibacterial activity of ELAVL1a.** The peptides E$_{1-180}$, E$_{181-242}$ and E$_{243-324}$ expressed in *E. coli* were purified by affinity chromatography on a Ni-nitrilotriacetic acid resin column. The purified recombinant peptides rE$_{1-180}$, rE$_{181-242}$ and rE$_{243-324}$ each yielded a single band of 23.74, 12.19, 13.00 kDa on SDS-PAGE after Coomassie Blue

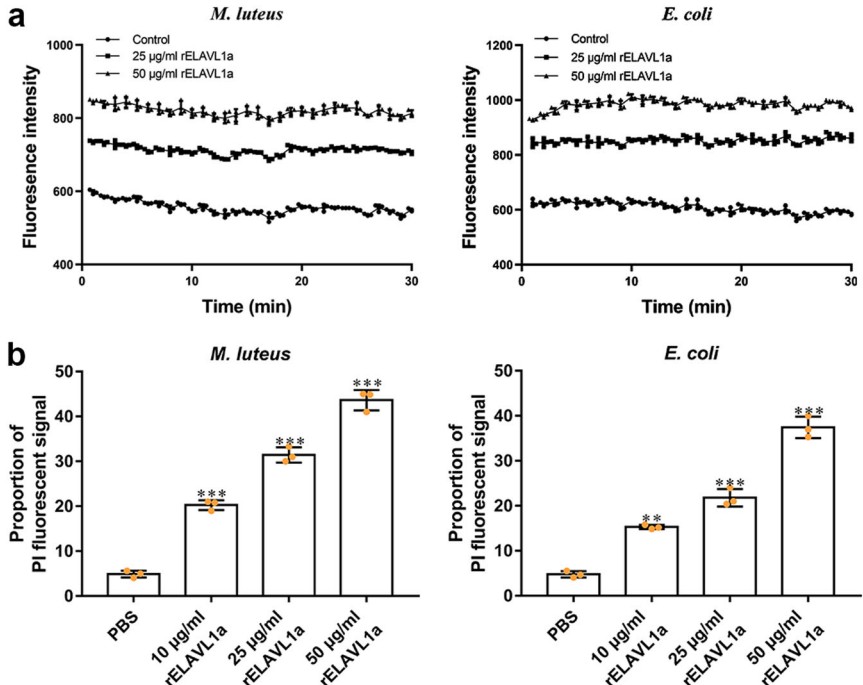

**Fig. 5 rELAVL1a was able to disrupt the bacterial membranes by a membrane lytic mechanism including a combined action of membrane depolarization and membrane permeabilization. a** The rELAVL1a caused depolarization of the bacterial plasma membrane. The changes in fluorescence intensity were recorded with a Tecan GENios plus spectrofluorometer at an excitation wavelength of 622 nm and an emission wavelength of 670 nm. Control, HEPES buffer containing 20 mM glucose. **b** The effects of rELAVL1a on the membrane integrity of *M. luteus* and *E. coli*, cells analyzed by flow cytometry. All data were expressed as mean values ± SD. ($n = 3$). The data are from three independent experiments performed in triplicate. The bars represent the mean ± S.D. The significance of the difference was determined by one-way ANOVA. $**p < 0.01$, $***p < 0.001$.

staining, matching the expected sizes, respectively (Fig.7a and Supplementary Fig. 15). Antibacterial assay showed that $rE_{181-242}$ still retained antibacterial activity against all the bacteria tested (Fig. 7b and Supplementary Fig. 16), but neither $rE_{1-180}$ nor $rE_{243-324}$ exhibited antibacterial activity against the bacteria (Supplementary Fig. 17). In accordance, $rE_{181-242}$ possessed affinity to LTA and LPS (Fig. 7c and Supplementary Fig.10c and 10d), but neither $rE_{1-180}$ nor $rE_{243-324}$ bound to the molecules (Fig. 7d, e, Supplementary Fig. 10e, f). Besides, the affinities of $rE_{181-242}$ as well as rELAVL1a to LTA or LPS were not inhibited by the presence of $E_{1-180}$ (Supplementary Fig. 10g–j), again indicating that like rELAVL1a, $rE_{181-242}$ could bind to LTA and LPS. It was clear that the antibacterial activity of $rE_{181-242}$ (and rELAVL1a as well) was closely related to its affinity to LTA and LPS. All these suggested that the C-terminal 181–242 residues of ELAVL1a comprising the linker between second RRM and third RRM were the core region indispensable for its antibacterial activity and affinity to LTA and LPS.

We also tested the antibacterial role of $rE_{181-242}$ in vivo. When 8- cell embryos were each microinjected with $rE_{1-180}$, $rE_{181-242}$, or $rE_{243-324}$, approximately 90% of the embryos developed normally. Challenge with live *A. hydrophila* caused an increase in the mortality rate of the embryos injected with PBS, BSA, $rE_{1-180}$, $rE_{243-324}$ or AcAb, with a 24-h cumulative mortality rate of ~67.3%, ~68%, ~67.3%, ~66.7%, and ~68.7%, respectively, while the same challenge induced only ~56% cumulative mortality at 24 h in the embryos injected with $rE_{181-242}$. Moreover, the embryo-protecting activity of $rE_{181-242}$ was counteracted by co-injection of ELAVL1aAb, but not by co-injection of AcAb (Fig. 7f). It was thus clear that $rE_{181-242}$, with antimicrobial activity in vitro, also showed antimicrobial activity in vivo, but $rE_{1-180}$ and $rE_{243-324}$, with no antibacterial activity in vitro, did not.

### Effects of amino acid replacement on antimicrobial activity.

Antimicrobial peptides are generally accepted to exert antimicrobial activities via their positively charged surface and amphipathicity. Thus, inverse PCR was used to generate mutants using the plasmid *pET28a/e181–242* as a template to examine the effects of charged and hydrophobic residue replacement on antimicrobial activity of the C-terminal 181–242 residues of ELAVL1a (Fig. 8a). All the mutants had a closely similar molecular mass (Fig. 8b), and we thus compared the antimicrobial activity of each mutant with that of control ($rE_{181-242}$) under the same concentration 50 μg/ml. Compared with $rE_{181-242}$, the double mutation V192G/I193S (m3; hydrophobic Val and Ile to hydrophilic Gly and Ser) and the double mutation R203G/R204G (m4; two consecutive positively charged Arg to neutral Gly) both resulted in a decrease in their antimicrobial activity against the bacteria tested (Fig. 8c and Supplementary Fig. 18) as well as their affinity to LTA and LPS (Fig. 8d). For two other mutants, the mutations V188G (m1; hydrophobic Val to hydrophilic Gly) and K189E (m2; positively charged Lys to negatively charged Glu), no conspicuous decrease in antimicrobial activity against the bacteria tested was observed. These suggested that the hydrophobic residues Val192/Ile193 as well as the positively charged residues Arg203/Arg204 were the important functional determinants contributing to the antimicrobial activity of rELAVL1a.

### Discussion

ELAVL1 (also known as HuR or HuA) has been shown to play multiple roles, including involvement in embryonic development, stem/progenitor cell survival, immune response, and tumorigenesis[15–22]. Previous studies on its function are mainly restricted to mammals, though ELAVL1a, a homolog of ELAV1 in zebrafish, has recently been reported to regulate erythropoiesis during development[28,29]. Overall, our understanding of the biological

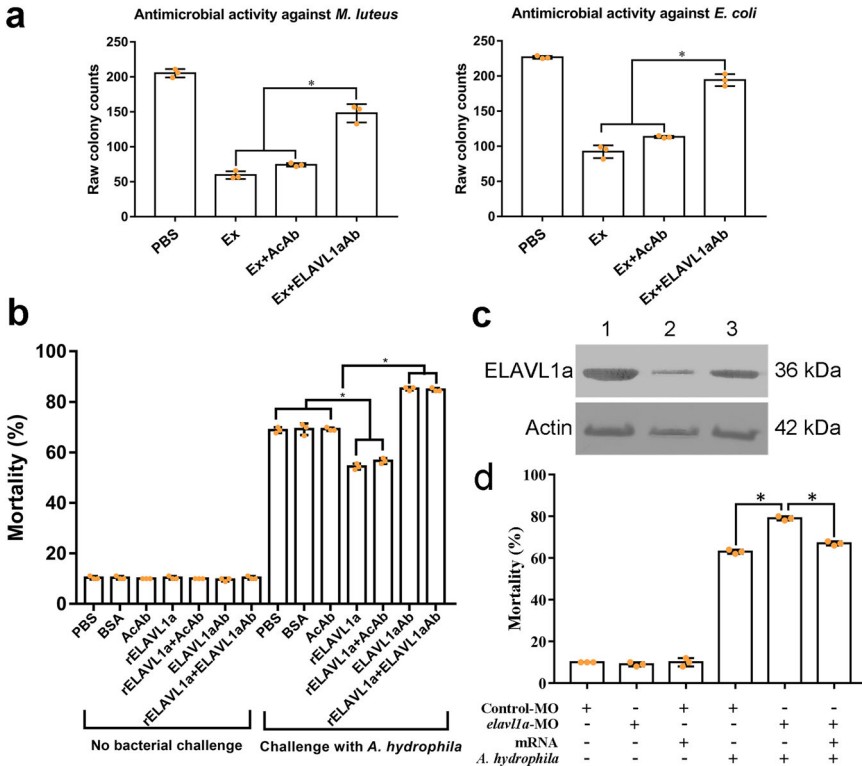

**Fig. 6 Antibacterial activity of ELAVL1a in vitro and in vivo. a** Antimicrobial activities of the embryo extract against *M. luteus* and *E. coli*. **b** The early (8-cell stage) embryos were first microinjected with PBS, BSA, anti-β-actin antibody (AcAb), anti-ELAVL1a antibody (ELAVL1aAb), rELAVL1a, and rELAVL1a plus AcAb or rELAVL1a plus ELAVL1aAb and then challenged by injection with live *A. hydrophila* ($8.3 \times 10^7$ cells/ml). The development of the embryos was observed, and the cumulative mortality rate was calculated at 24 h after bacterial injection. **c** Western blotting analysis of the efficacy of *elavl1a*-MO. β-Actin was used as control. Lane 1, 2, 3, the levels of ELAVL1a and β-Actin in the embryos injected with control morpholino (1), *elavl1a*-MO (2), or co-injection of *elavl1a*-MO plus *elavl1a*-mRNA (3). **d** The reduced antibacterial activity of the *elavl1a*-MO-knockdown embryos. The 24-h mortality of *elavl1a*-MO-knockdown group, control group, and rescued group, after the challenge with live *A. hydrophila* ($8.3 \times 10^7$ cells/ml). All data were expressed as mean values ± S.D. ($n = 3$). The data are from three independent experiments performed in triplicate. The bars represent the mean ± SD. The significance of the difference was determined by one-way ANOVA. *$p < 0.05$. M, marker; con, control; Ex, embryo extract; AcAb, anti-β-actin antibody; ELAVL1aAb, mouse anti-ELAVL1a antibody.

functions of ELAV1 is rather poor and incomprehensive, especially for non-mammalian vertebrates. Here we demonstrate that ELAVL1a is not only as a pattern recognition receptor, capable of identifying LTA and LPS as well as bacteria, but also an effector molecule, capable of suppressing the growth of Gram-positive and negative bacteria. The binding of ELAVL1a to Gram-positive and -negative bacteria are apparently through interaction with LTA and LPS, respectively. Potential mechanisms of antibacterial proteins include interaction with or insertion into bacterial membrane, which cause either fatal depolarization of the normally polarized membrane, or formation of physical pores, or scrambling of the usual distribution of lipids between the leaflets of the bilayer or damage to critical intracellular targets. We show that ELAVL1a functions by a combined action of membrane lytic mechanisms including interaction with bacterial membrane via LTA and LPS (especially lipid A), membrane depolarization and membrane permeabilization. We also identify ELAVL1a as an uncharacterized maternal immunocompetent factor. Evidences supporting this includes abundant storage of ELAVL1a protein in eggs and embryos; correlation between ELAVL1a and embryonic antibacterial capacity; and marked reduction of embryonic antibacterial activity by knockdown with *elavl1a*-MO. In addition, we prove that zebrafish egg/embryo contains enough ELAVL1a to exert antibacterial activity in vivo. Collectively, these denote that ELAVL1a is a maternal factor functioning in the egg/embryo of zebrafish. This apparently has an important physiological

implication. In contrast to development of mammalian embryos in the uterus inside the mother's body, most fish embryos are exposed to a hostile aquatic environment full of potential pathogens capable of causing various types of diseases. They are thus very fragile and vulnerable to attack by pathogens. ELAVL1a, as a maternal immunocompetent factor, can protect fish embryos in vivo against the attacks by pathogens like *A. hydrophila*. However, it needs to point out that ELAVL1a may function only when the pathogens have penetrated the embryos, as it is localized in the cytoplasm.

Fishes are the basal vertebrates whose immune system has been shown comparable with that of mammals. Currently, we do not know if ELAV1 of mammalian species has any antibacterial activity, though HuR/HuA has been shown involved in immune response[20–27]. Given the high conservation of ELAVL1 from fish to mammals, we speculate that mammalian ELAVL1 may play a similar antibacterial role in early development. However, this needs further study in the future.

Proteins are built as amino acid chains, which are tied to functions. We prove that the C-terminal 62 residues of ELAVL1a, positioned at 181–242, are the critical sequence contributing to the antibacterial activity of ELAVL1a. We also prove that the antibacterial activity against the Gram-positive and -negative bacteria and the affinity to LTA and LPS co-exist in both $rE_{181-242}$ and rELAVL1a, but neither the antibacterial activity nor the affinity to LTA and LPS are present in $rE_{1-180}$ and $rE_{243-324}$,

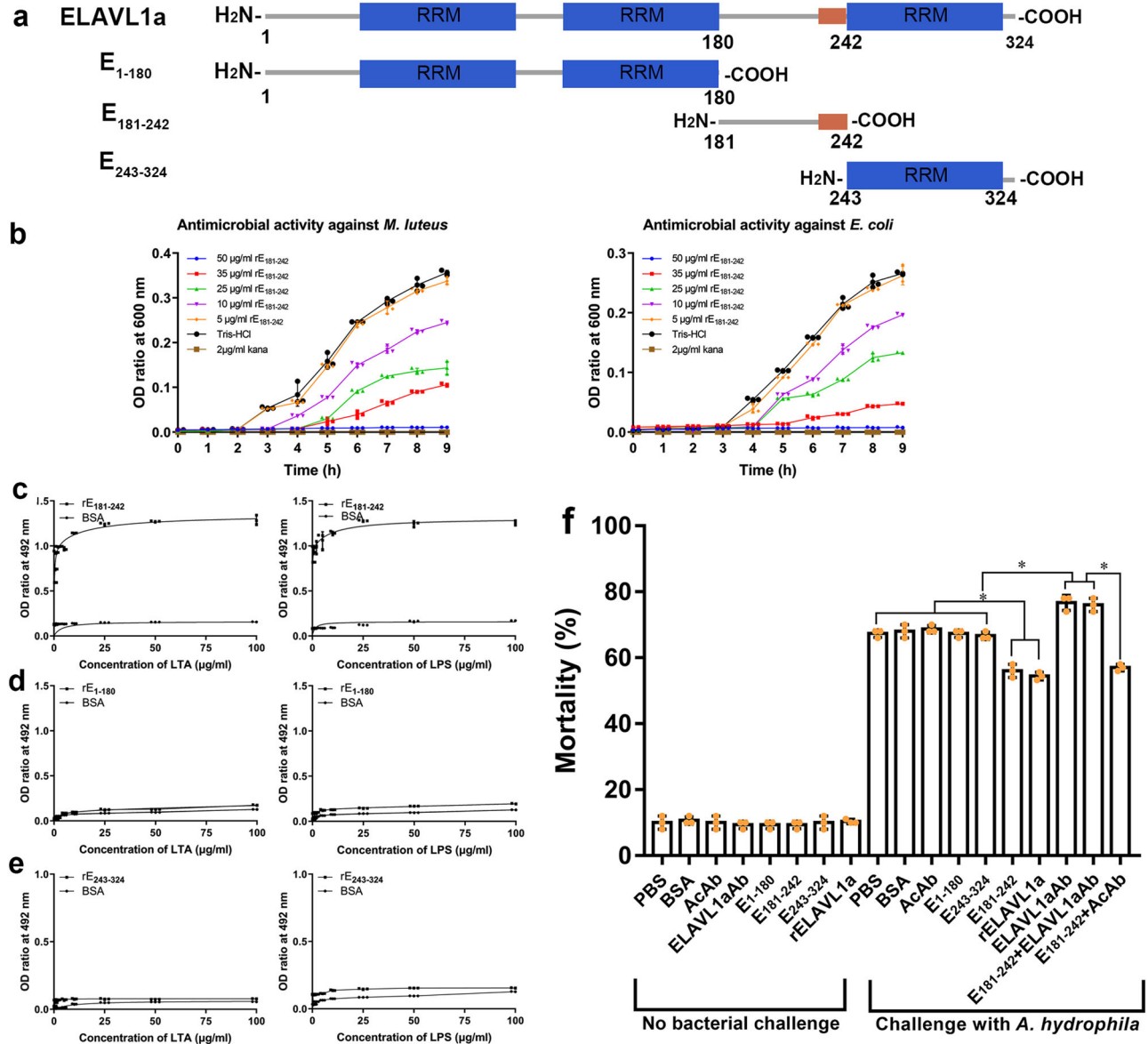

**Fig. 7 Diagram showing zebrafish ELAVL1a truncation and their antibacterial activities and binding to LPS and LTA. a** Diagram showing zebrafish ELAVL1a truncation. **b** Antibacterial activity of rE$_{181-242}$ against *M. luteus* and *E. coli*. **c, d, e** Interaction of rE$_{181-242}$ (**c**), rE$_{1-180}$ (**d**) and rE$_{243-324}$ (**e**) with LTA and LPS revealed by ELISA. **f** The in vivo bioactivity of rE$_{1-180}$, rE$_{181-242}$, and rE$_{243-324}$. The early (8-cell stage) embryos were first microinjected with PBS, BSA, anti-β-actin antibody (AcAb), anti ELAVL1a antibody (ELAVL1aAb), rE$_{1-180}$, rE$_{181-242}$, rE$_{243-324}$ and rELAVL1a, then challenged by injection with live *A. hydrophila*. The development of the embryos was observed and the cumulative mortality rate was calculated at 24 h after injection. All data were expressed as mean values ± S.D. (*n* = 3). The data are from three independent experiments performed in triplicate. The bars represent the mean ± SD. The significance of the difference was determined by one-way ANOVA. *$p < 0.05$. AcAb, anti-β-actin antibody; ELAVL1aAb, mouse anti-ELAVL1a antibody.

suggesting a correlation between antibacterial activity and LTA- and LPS-binding activity. Probably, this correlation forms the basis for the functional sites of rELAVL1a and rE$_{181-242}$ simultaneously involved in multiple activities, including interacting with the bacterial signature molecules LTA and LPS and destabilizing/disrupting the bacterial cell membranes.

It is known that most antimicrobial peptides are cationic and amphiphilic molecules with broad-spectrum antimicrobial activity against a wide range of bacteria. We show by side-directed mutagenesis that the hydrophobic residues Val192/Ile193 and the positively charged residue Arg203/Arg204 of the C-terminal 62 residues are the functional determinants contributing to the antimicrobial activity of ELAVL1a, whereas neither the hydrophobic Val188 nor the positively charged Lys189 had little contribution to its

antimicrobial activity. This suggests that both the position and distribution of hydrophobic and positively charged residues are the major factors mediating the specific electrostatic surface and amphipathicity of the C-terminal 62 residues of ELAVL1a.

In summary, this study highlights the identification of ELAVL1a as a maternal immunocompetent factor which can protect the embryos of zebrafish via recognizing LTA and LPS as well as bacteria and killing the Gram-positive and -negative bacteria through interacting with them and disrupting their plasma membranes. Our study also reveals that the C-terminal 62 residues are a key region for the antibacterial activity of ELAVL1a. This work provides an additional viewpoint for understanding the biological roles of ELAVL1 proteins, which are ubiquitously present in metazoan.

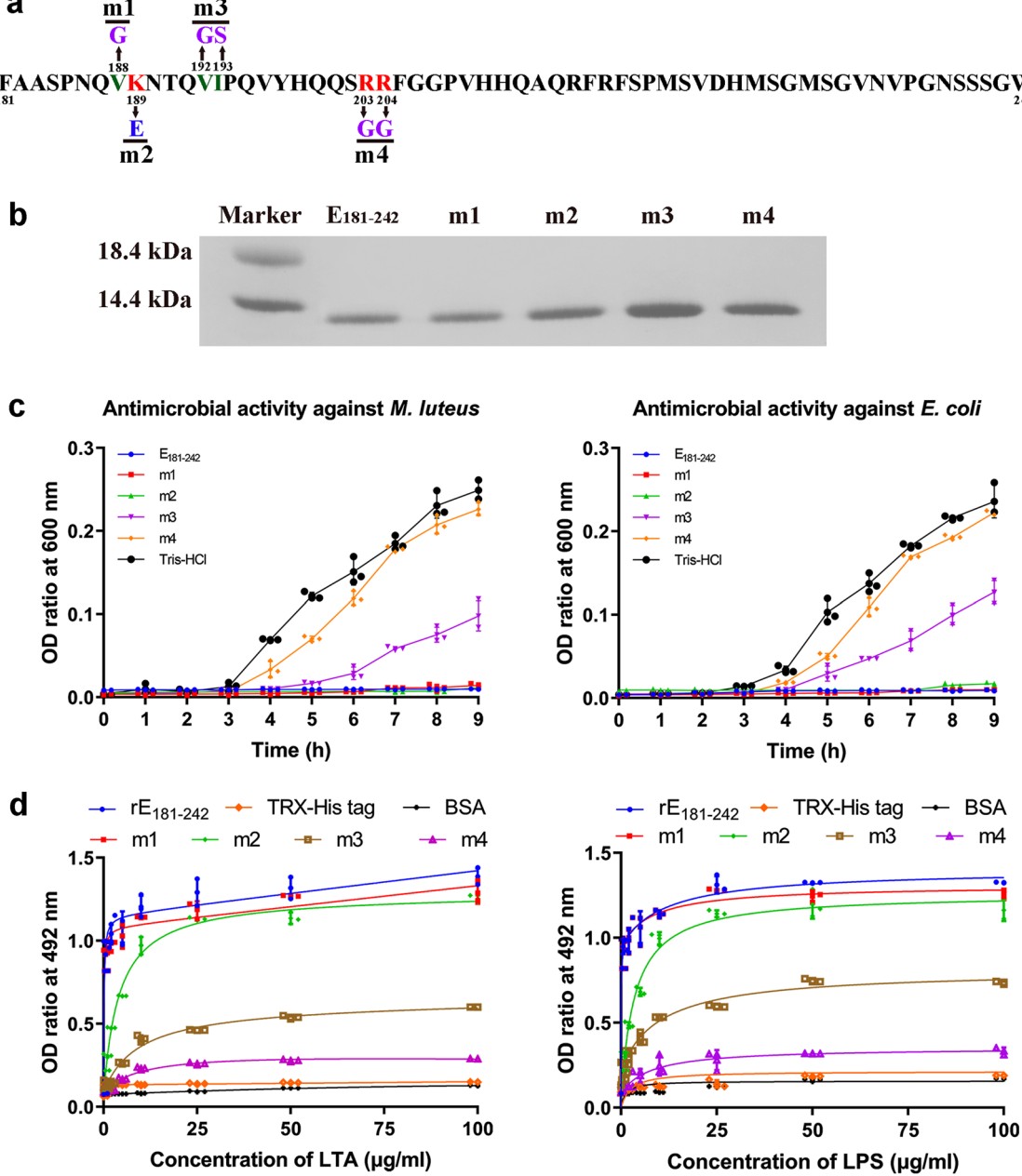

**Fig. 8 Antibacterial activity of rE$_{181-242}$ mutants. a** Diagram showing the site-directed mutagenesis. **b** SDS-PAGE of the mutants, showing that they all have a similar molecular mass. **c** Antibacterial activities of rE$_{181-242}$ mutants against *M. luteus* and *E. coli*. rE$_{181-242}$ was used as positive control and Tris-HCl buffer used as negative control. **d** Interaction of E$_{1-180}$ mutants with LTA and LPS revealed by ELISA. Each point in the graph represents the mean ± SD ($n$ = 3). The data are from three independent experiments performed in triplicate. The bars represent the mean ± SD.

## Methods

**Fish and embryos**. All the fish used were treated in accordance with the guidelines of the Laboratory Animal Administration Law of China with the permit number SD2007695 approved by the Ethics Committee of the Laboratory Animal Administration of Shandong province. The wild-type zebrafish *D. rerio* purchased from a local fish dealer were maintained as described by Wang et al.[32]. Sexually mature *D. rerio* were placed in a container in the late evening at a female to male ratio of 2:1, and the naturally fertilized eggs were collected in the next early morning and cultured at 27 ± 1 °C until use.

**Western blotting**. Our previous studies have revealed ELAVL1a being an LTA-binding protein[30]. To examine the distribution of ELAVL1a in the different tissues and at the different developmental stages, the tissues including the heart, liver, spleen, gut, kidney, muscle, skin, brain, eye, gill, ovary, testis, and tail were dissected out of zebrafish, and the embryos/larvae including newly fertilized eggs, 4-, 16-, and 256-cell embryos, high blastulae and gastrulae, 10-somite larvae, 2-day-old larvae, 3-day-old larvae, and 7-day-old larvae collected. All the samples were

homogenized in PBS (pH 7.4) using a JY92-IIN sonicator (Scientz, Ningbo, Zhejiang, China), and the homogenates were centrifuged at 5000 × *g* at 4 °C for 10 min. The protein concentration of the supernatants was determined using a Bicinchoninic Acid Protein Assay Kit (ComWin Biotech, Beijing, China). The supernatants were loaded onto and run on a 12% SDS-PAGE gel. Western blotting was performed according to the method of Wang et al.[32]. The primary antibody used was mouse anti-human ELAVL1 antibody (ABIN577055; Clonegene, Hartford, Connecticut, USA), which was diluted 1:1000 with PBS (pH7.4). Unprocessed original blot images are provided in Supplementary Figs. 19 and 20.

**Immunohistochemistry**. The localization of ELAV1a in embryonic cells was detected by the method of Wang et al.[32], using mouse anti-human ELAVL1 antibody diluted 1:500, and mouse IgG1 isotype control antibody (ABIN457406; Clonegene, Hartford, Connecticut, USA) as control. To stain nuclei, the embryos were counterstained with 5 µg/ml 4′,6-diamidino-2-phenylindole (DAPI; Solarbio, Beijing, China) in PBS for 10 min, washed in PBST for 10 min, and stored at 4 °C.

The embryos/larvae were observed and photographed under a Leica confocal microscope (Leica, Wetzlar, Hessen, Germany).

**Cloning and sequencing of *elavl1a*.** Total RNAs were isolated from 60 embryos at about the 128-cell stage. The first-strand cDNAs were synthesized with a reverse transcription kit (TaKaRa, Otsu, Shiga, Japan) with an oligo (dT) primer and stored at −20 °C till used.

A pair of the primers P1 and P2 (Supplementary Data 1) specific of ELAVL1a gene was designed using the Primer Premier program, version5.0, based on the sequence of *elavl1a* (NM_131452.1) on NCBI and used for PCR. The amplification product was cloned into pGEM-T vector (Tiangen Biotech, Beijing, China) according to the manufacturer's instructions, and then transformed into Trans 5α bacteria (Tiangen Biotech, Beijing, China). The positive clones were selected and sequenced by Shanghai Sangon Biotech (China) to verify for authenticity.

**Homology and phylogenetic analyses.** The cDNA obtained was analyzed for coding probability with DNASTAR software package (version 5.0), and the domain and signal peptide were predicted using SMART program (http://smart.embl-heidelberg.de/) and the SignalP 4.1 Server (http://www.cbs.dtu.dk/services/SignalP/), respectively. The molecular mass (MW) and isoelectronic point (pI) of the mature protein were determined using ProtParam (http://www.expasy.ch/tools/protparam.html). The nuclear localization signal was analyzed by PSORT II (http://psort.hgc.jp/form2.html). Homology search in GenBank database was carried out by BLAST server (http://www.ncbi.nlm.nih.gov/BLAST/), and phylogenetic tree constructed by MEGA (version 7.0) using p-distance based on neighbor-joining method. The reliability of each node was estimated by bootstrapping with 1000 replications. The numbers shown at each node indicate the bootstrap values (%). The three-dimensional (3D) structure prediction was performed by SWISS-MODEL online software at Expert Protein Analysis System (http://www.expasy.org/) using human ELAVL1 (PDB code: 4egl.1.A) as model.

**Quantitative real-time PCR.** Quantitative real-time PCR was used to examine the expression profiles of *elav1a* in both different tissues and different stage embryos as described by Wang et al.[32]. The primers P3 and P4 (Supplementary Data 1) specific of *elavl1a* were designed using the Primer Premier program, version 5.0. The gene *β-actin* (P5 and P6; Supplementary Data 1) was chosen as the reference for internal standardization.

**Whole-mount in situ hybridization.** The fertilized eggs, 2-cell embryos, high blastulae and gastrulae, 10-somite larvae, and 1-, 2-, and 3-day-old larvae were collected. A fragment of *elavl1a* was PCR-amplified using the primer pair P7 and P8 (Supplementary Data 1) and subcloned into vector pGEM-T (Tiangen Biotech, Beijing, China). Digoxigenin (DIG)-labeled *elavl1a*-specific antisense riboprobes were synthesized with linearized vectors (digested by *Nco* I enzyme) and Sp6 RNA polymerase (Thermo Fisher Scientific, Waltham, MA, USA) through in vitro transcription. Whole-mount in situ hybridization was performed by the method of Thisse and Thisse[33]. After staining, the embryos/larvae were observed and photographed under Nikon SMZ1000 stereomicroscope.

**Expression and purification of recombinant ELAVL1a.** The cDNA encoding complete ELAVL1a was amplified by PCR using the sense primer P9 (*EcoR* I site is underlined) and the antisense primer P10 (*Hind* III site is underlined) (Supplementary Data 1). The PCR product was digested with *EcoR* I and *Hind* III, subcloned into pET28a expression vector (Novagen, Darmstadt, Hessen, Germany) digested previously with the same restriction enzymes, and the plasmid designated pET28a/elavl1a. The cells of *Escherichia coli* BL21 were transformed with the plasmid pET28a/elavl1a and then processed as described by Du et al.[34]. To express thioredoxin-histidine-tag (TRX-His-tag) peptide for control, *E. coli* BL21 cells were also transformed by plasmid pET32a. The induction and purification of recombinant TRX-His-tag peptide were performed as described above. Protein concentrations were determined by BCA protein assay kit (CWBIO, Beijing, China), using bovine serum albumin as a standard.

**Assay for affinity of ELAVL1a to bacteria, LTA, LPS, and lipid A.** The Gram-positive bacteria *Micrococcus luteus* (ATCC 49732), *Bacillus subtilis* (ATCC 6633) and *Staphylococcus aureus* (ATCC 25923) and the Gram-negative bacteria *E. coli* (ATCC 25922), *Vibrio anguillarum* (ATCC 43308) were incubated at 37 °C for 16 h. The Gram-negative bacterium *Aeromonas hydrophila* (ATCC 35654) was incubated at 28 °C in Tryptic Soy Broth (TSB) medium for 16 h. All the bacteria were collected by centrifugation at $5000 \times g$ at room temperature for 10 min, re-suspended in PBS, yielding a concentration of $1 \times 10^8$ cells/ml. The binding of recombinant ELAVL1a (rELAV1a) to bacteria, LTA, LPS, and lipid A was assayed as described by Wang et al.[32]. The binding of rELAVL1a to LTA and LPS was also analyzed by the pulldown experiments. In brief, 100 μg/ml rELAVL1a or recombinant TRX-His tag was loaded onto an LTA- or LPS-conjugated resin column and incubated at 4 °C overnight. The column was eluted with at least 10 volumes of the initial buffer (10 mM Tris–HCl with 150 mM NaCl, pH7.4) to remove the unbound proteins, and then with the elution buffer (4 M Urea in 10 mM Tris–HCl, pH7.4) to recover the bound

proteins from the affinity matrix. All the effluent fractions were separated by SDS-PAGE. To test if ELAVL1a has any lectin activity, the binding of rELAV1a to LPS was also tested in the presence of the sugars D-glucose, D-galactose, D-mannose, GlcNAc, N-acetyl-D-galactosamine (GalNAc), and Nacetyl-D-mannosamine (0, 2.5, 5, 10, and 20 mg/ml) as described by Gao et al.[35]. For the competition assay, rELAVL1a (25 μg/ml) was pre-incubated with LTA or LPS, and then processed as above.

**Assay for antibacterial activity of ELAVL1a in vitro.** The antibacterial activity of ELAVL1a in vitro was measured as described by Du et al.[34] with slight modifications. Briefly, the Gram-positive bacteria *M. luteus, B. subtilis,* and *S. aureus* and the Gram-negative bacteria *E. coli, V. anguillarum, A. hydrophila,* and *Edwardsiella tarda* (an intracellular bacterium; EIB202, from Marine Culture Collection, Ocean University of China, MCCO) were each incubated with different concentrations of rELAVL1a at 37 °C, and inhibition of the bacterial growth was determined by measuring absorbance at 600 nm with Multiskan MK3 microplate reader (Thermo Fisher Scientific, MA, USA). For the Oxford cup agar diffusion, aliquots of 100 μl of 20 mM Tris–HCl buffer (pH7.4) containing 25, 35, and 50 μg/ml rELAVL1a were each added to an Oxford cup and then placed on agar plates containing $10^5$ cells of tested bacteria. The blank and negative controls were processed similarly, except that the rELAVL1a was replaced by 20 mM Tris–HCl buffer or 20 mM Tris–HCl buffer containing 50 μg/ml recombinant TRX-His tag peptide.

**Transmission electron microscopy.** Transmission electron microscopy (TEM) was used to detect the bactericidal activity of rELAVL1a, as described by Liu et al.[36].

**Assay for membrane depolarization and permeabilization activities of rELAVL1a.** The depolarization and permeabilization activities of rELAVL1a towards bacterial membranes were measured, as described by Du et al.[34].

**Assay for antibacterial activity of ELAVL1a in embryos.** The antibacterial activity of embryo extracts against the Gram-positive bacteria *M. luteus, B. subtilis,* and *S. aureus* and the Gram-negative bacteria *E. coli, V. anguillarum,* and *A. hydrophila* was detected by the method of Wang et al.[37]. The antibacterial activity of rELAVL1a in vivo was assayed as described by Du et al.[34]. with slight modifications. In brief, 8-cell stage embryos were each microinjected into the yolk sac with 6 nl of sterilized PBS containing 0.33 ng of mouse anti-ELAVL1a antibody, 0.33 ng of anti-β-actin antibody, 0.6 ng of purified rELAVL1a, 0.6 ng of BSA, 0.6 ng purified rELAVL1a plus 0.33 ng of mouse anti-ELAVL1a antibody or 0.6 ng purified rELAVL1a plus 0.33 ng of anti-β-actin antibody, and then challenged 1 h later by injection of 6 nl of live *A. hydrophila* (pathogenic to zebrafish) suspension containing about 500 bacterial cells. For each group above, a total of 50 embryos were microinjected. The mortality was recorded, and the cumulative mortality calculated at 24 h after the bacterial injection.

**Assay for antibacterial activity of *elavl1a*-MO-knockdown embryos.** The sequence-specific antisense oligonucleotides of 25 nucleotide, TGTGGTCTT CGTAACCGTTCGACAT (morpholino for *elavl1a*, *elavl1a*-MO) and CCTC TTACCTCAGTTACAATTTATA (control morpholino, control-MO) were designed and synthesized by Gene Tools, LLC (Philomath, OR, USA). One hundred of 1- to 2-cell stage embryos were microinjected with 12 ng/6 nl of *elavl1a*-MO. Half of the embryos were challenged 24 h later by injection of 6 nl of live *A. hydrophila* suspension (~500 cells). For control, the embryos were injected with the control-MO and treated similarly. The efficacy of *elavl1a*-MO was confirmed by Western blotting. For the rescue experiment, the open reading frame (ORF) of *elavl1a* was cloned into vector pCS2, and ORF mRNA was synthesized by mMACHINE mMESSAGE SP6 Kit (Life Technologies, Carlsbad, CA). Acquired ORF mRNA was co-injected with *elavl1a*-MO. All the embryos were cultured at 27 °C and the mortality was evaluated 24 h later after the challenge with *A. hydrophila*.

**Titration of ELAVL1a content in eggs/embryos.** Exactly, 120 embryos collected at 0, 12, and 24 hpf were washed three times with sterilized 0.9% saline, homogenized, and centrifuged at $5000 \times g$ at 4 °C for 5 min. The supernatants were pooled and used to measure the content of ELAVL1a by ELISA as described by Wang et al.[32].

**Assay for structure-activity relationship.** To determine structure-activity relationship, the cDNA regions encoding the N-terminal 180 residues containing two RRMs ($E_{1-180}$; with the C-terminal 144 residues depleted), the C-terminal 62 residues at 181–242 consisting of the linker region between second RRM and third RRM ($E_{181-242}$) and the C-terminal 82 residues 243–324 ($E_{243-324}$; with the N-terminal 242 residues depleted) of ELAVL1a were amplified by PCR using the pairs of primers P11 and P12, P13, and P14, as well as P15 and P16 (Supplementary Data 1). The expression vector plasmids pET28a/e1–180, pET28a/e181–242, and pET28a/e243–324 were constructed and transformed into *E. coli* BL21 cells. The expression and purification of recombinant proteins were performed as above, and

the purified proteins verified by Western blotting using anti-His-Tag mouse monoclonal antibody. The antibacterial activities of $rE_{1-180}$, $rE_{181-242}$, and $rE_{243-324}$ against the Gram-positive and Gram-negative bacteria as well as their affinity to LTA and LPS were both assayed as above. The pull-down and competition binding assays between $rE_{1-180}$ (and rELAVL1a) or $rE_{181-242}$ and LTA or LPS were performed as described above. In addition, $rE_{1-180}$, $rE_{181-242}$, and $rE_{243-324}$ were also microinjected into zebrafish embryos, followed by the challenge of live *A. hydrophila*, to test their antibacterial activity in vivo.

**Site-directed mutation of ELAVL1a and bioactivity assay**. The plasmid *pET28a/e181–242* was used for mutational analyses. Mutants were generated using Mut Express II Fast Mutagenesis Kit V2 (Vazyme, Nanjing, China) according to the kit's protocol. The specific primers used are listed in Supplementary Data 1. The mutations were confirmed by DNA sequencing. Expression and purification of the mutated recombinant proteins and bioactivity assay were carried out as described above.

**Statistics and reproducibility**. All the assays were performed in triplicate (technical replicates), and each experiment was repeated three times (biological replicates). Statistical analyses were performed using GraphPad Prism 5. The band intensity of western blot was quantified by ImageJ software. The significance of difference was determined by one-way ANOVA. $p$-values of 0.05 or less were considered statistical significance ($*p < 0.05$, $**p < 0.01$, $***p < 0.001$) and shown in figures. All of the data were expressed as mean ± SD. Source data underlying the graphs are provided in Supplementary Data 2.

**Reporting summary**. Further information on research design is available in the Nature Research Reporting Summary linked to this article.

## Data availability

All data that support the findings of this study are available from the corresponding author upon request. Source data underlying the graphs are shown in Supplementary Data 2.

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

## Acknowledgements

We are grateful to Wei Yang (Institute of Evolution and Marine Biodiversity, Ocean University of China) for the technical assistance with flow cytometry and to Xiaoyu Wang (Institute of Evolution and Marine Biodiversity, Ocean University of China) for aid in immunohistochemistry. This work was supported by the National Natural Science Foundation of China (32073000) and the Ministry of Science and Technology (MOST) of China (2018YFD0900505).

## Author contributions

Z.S.C. and N.S.S. designed the experiments and wrote the paper. N.S.S., Z.Y., S.L.L., C.Y., D.X.Y., and W.X. performed the experiments. N.S.S. and Z.S.C. analyzed the data. All authors approved the final version of the manuscript.

## Competing interests

The authors declare no competing interests.
