## [Peer Review File · Communications Biology]

Reviewers' comments:

Reviewer #1 (Remarks to the Author):

This work shows potentially biologic functional roles of ELAVL1a in zebrafish. Authors found that ELAVL1a is a LTA- and LPS-binding protein, capable of inhibiting the growth of various Gram-positive and -negative bacteria and explored the possible mechanism underlying the disruption of bacteria, including membrane depolarization and membrane permeabilization. Then they found that C-terminal 62 residues positioned at 181-242 of ELAVL1a is critical for the antibacterial activity. In addition, authors tested the antibacterial activity of ELAVL1a in vitro and in vivo in an attempt to try to confirm the results. However, many results in this work need to be validated, and the issues the authors should attend to as listed below.

Major concerns

1. Since the antibody used in the in vivo blockade assay was a mouse anti-human ELAVL1a antibody, it is important for authors to validate if this antibody works correctly in zebrafish, including the specificity and efficiency of this antibody for a zebrafish ELAVL1a homolog. Western blot assay is recommended for this procedure, in which the recombinant zebrafish ELAVL1a protein (rELAVL1a) and particularly the natural ELAVL1a proteins from zebrafish tissues and embryos should be included.
2. The binding activity of zebrafish ELAVL1a protein to LTA, LPS and lipid A is the main discovery of the study. The authors claim that both LTA and LPS were recognized by ELAVL1a through a common C-terminal 62 residues positioned at 181-242, but in view of the large differences in composition and structure between LTA and LPS/lipid A molecules, it looks likely that nonspecific interactions may exist between ELAVL1a and LTA or LPS/lipid A. More solid evidence is needed to further exclude the possible nonspecific interactions and support the author's conclusion. It is recommended that various immunoprecipitation assays in different combinations should be added, including immunoprecipitation between the wild type ELAVL1a protein and LTA/LPS/lipid A, as well as between the mutant ELAVL1a proteins (deletion with C-terminal 62 residues and N/C-terminal RRM domains) and LTA/LPS/lipid A. Competitive binding assays should be included in the immunoprecipitation by introducing mutant ELAVL1a proteins, such as introducing the mutant ELAVL1a protein with deletion of C-terminal 62 residues in the experiment between the wild type ELAVL1a protein and LTA/LPS/lipid A. In addition, clarification of the detailed functional residues in the C-terminal 62-aa segment that contribute to the interaction of ELAVL1a with LTA/LPS/lipid A by charged and/or hydrophobic interactions will be benefit to support the conclusion.
3. For bactericidal examination, plaque inhibition assay should be included in the study in addition to the growth inhibition test.
4. Subcellular localization analysis showed that the ELAVL1a protein distributes in the nucleus and cytoplasm, thus an intracellular bacterial infection model should be used instead of extracellular bacterial infection.
5. In figure 2A, B, essential isotype antibody control data is missing. Please included these data in a supplementary figure, which is critical to exclude the non-specific binding of antibody to the zebrafish cells.
6. In figure 5E, F the image is blurry. Please show high-quality merged images for the bacteria with FITC-labeled ELAVL1a. In addition, in figure 5G-L, results for the binding of TRX-His peptide to LTA, LPS.....are missing.
7. Again, in figure 6A, essential control data is missing. Since the ELAVL1a is fused with TRX-His tag, it is critical to add the control experiments to exclude the effect of TRX-His tag. Or you can cut the TRX-His tag with corresponding enzyme.

8. In figure 8C, authors should add the rescue experiment to confirm the specificity of MO.

Minor concern

9. In figure 7A, what does "control" specifically mean?

10. English writing needs to be improved.

11. Authors described ELAVL1a as a pattern recognition receptor in abstract and discussion sections, there is not enough evidence for this conclusion.

Reviewer #2 (Remarks to the Author):

Brief summary:

The current study talks about identification and characterization of zebrafish ELAVL1a which is an analog of mammalian HuR. ELAVL/Hu are a family of RNA binding proteins which regulate gene expression at post transcriptional level and are necessary for proper embryonic development. With a robust analysis of experiments, they go on to demonstrate that a function of ELAVL1A is to provide immunity from bacteria during embryonic development. The authors state and prove that ELAVL1a binds to LPS and LTA specifically on Gram-negative and Gram-positive bacteria and disrupts the membrane via depolarization and permeabilization. They also demonstrate that the C-terminal 62 residues from position 181-242, which does not contain any RRM, were vital in providing this anti-bacterial activity. Furthermore, inhibition of ELAVL1a via antibody or gene knockdown resulted in increased mortality as the pathogen resistant activity was reduced. Collectively, this work shows that ELAVL1 is an immune factor in zebrafish embryos that can protect from bacterial infection.

Overall impression of the work:

The authors of this paper have done a fine job of combining computational sequencing and modelling with protein purification and came up with a robust body of work which can influence others in its stead. The quality of data is adequate in most cases accompanied with appropriate controls. There is very little room for criticism on the amount of work done and no need for additional experiments to be done to corroborate their findings. However, there is a significant amount of data that should be moved to the supplement or completely removed from the paper. Also, certain data need to be quantified in order to be presented and the statistical tests used must be clearly named in the legend so that the reader may determine their appropriateness. There are also several improvements that can be made to the way the article is organized and formatted. The authors should carefully proofread their work and change some figures to make it easier for the reader to follow. The following comments on the manuscript should help to guide them in improving their study.

This work will provide more context to the previously undefined role of HuR family of proteins in providing immunity during embryonic development.

Major Concerns

Figure 1.

- It is concerning that the sequence coverage for the protein ID of band 5 was only 7% of the ELAVL1a sequence. This may be because it seemed to be one of the minor bands. Was there a control for non-specific binding done in this experiment? It would be best to show pull down result using un-conjugated Sepharose as a control.
- There is a lot of wasted space in this figure. I would suggest rearranging by placing the sequence

coverage information in the text or legend and including to Fig 3a and 3b panels in this figure. The flow should be as such: 1a. pull-down gel, 1b. ELAV1a domain structure, 1c. ELAV1a 3D structure prediction, 1d. organ blots, 1e. stage blots. The phylogenetic tree 3c. can be moved to the supplement.

- Actin bands look not consistent. Especially in b, it looks underexposed. The quantification of the Western blot can just be numbers below the band. That way it is easier to interpret it rather than in graphical format.

- Strike the statement in line 74-75 that ELAVL1a is relatively higher in the ovary. The bands for heart, ovary and testes are all oversaturated so differences in abundance in these organs is impossible to determine from these data.

Figure 2.

- 2a: the green signal in the 488 channel is completely oversaturated

- Figure 2b, the green and blue (Alexa 488 and DAPI) fluorescence looks inconsistent between A, B and C as if taken at different exposures.

- Fig 2b and 2c: without quantification this data is not convincing. I do not see strong evidence of nuclear localization in these figures. These two panels are unnecessary for the main conclusions of the paper so I would suggest removing them entirely unless quantification data is available.

- After removing the localization data in 2b and 2c, I would suggest combining 2a with figure 4 to make a complete figure 2 about the expression of ELAV1a in different stages and tissues.

Figure 3.

- Combine into figure 1 as suggested above

Figure 4.

- Combine into figure 2 as suggested above

- For y axis legends change to Relative ELAV1a expression

- Figure 4c: WISH color seems diffused and not specific. Is it the blue that we're supposed to focus on or the purple? Not clear.

Figure 5 (Now Figure 3)

- Figure 5e and f: The fluorescence and the scale bars do not seem very clear. You can barely see the bacteria. It would be better if the images were zoomed in and the bacteria were clearly visible (100X magnification). Also, these data should be quantified to be impactful. If there is no quantification, this data needs to be moved to the supplement.

Figure 7 (Now Figure 5)

- Figure 7a: The experimental set up was not clearly explained in the main text or legend. What is the fluorescence intensity being measured? How does it demonstrate loss of membrane polarization? Please provide references.

- Indicate the statistical test used to determine p values in the legend.

Figure 8 (Now Figure 6)

- Indicate the statistical test used in the figure legend and show error in both directions

- What was the MOI of the challenge with *A. hydrophilia*? This needs to be indicated somewhere in the legend. It appears to be too high as still over 50% of embryos die even with injection of exogenous ELAVL1a.

- The experiment in 8d. is an odd way to show that the bacteria were being killed. This is normally done by CFU assay as bacterial DNA can be present even when no viable bacteria are present. There is also no loading control. This data should be moved to the supplement or removed entirely.

Figure 9 (Now Figure 7).

- Panel B should be moved to the supplement.

- Please indicate the statistical test used in the legend.

Minor concerns:

1. For all graphs error bars need to be shown in both directions.

2. Line 23: should be biological instead of biologic

3. Line 228: Here instead of Her

4. Line 296: verify instead of verity

5. Figure 2a and b: Alexa 488 should be renamed to ELAVL1a as that's what we're observing rather than the channel.
6. Figure 5a and d: These can be moved to supplement.
7. Figure 5a, b, c and d: Labelling them appropriately in the figure rather than in figure legend would be clearer. Label blots above with +/- for the added ELAV1a and the bacteria input instead of 1-7.
8. Line 569: Gram-positive instead of Gran-positive.
9. Figure 6a: Some graphs could be moved to supplement. Just one bacterium from Gram-positive side and one from Gram-negative side would be enough.
10. Figure 7 caption: membranolytic is not a word.
11. Figure 7a and b: Same as comment 9.
12. Figure 8a: Same as comment 9.
13. Figure 9c: Same as comment 9.
14. Known functions of mammalian HuR protein are quite different from ELAVL1a as listed in this paper. It would be helpful to know if Zebrafish ELAVL1a has similar functions to its mammalian counterpart

Response to referees

To Reviewer #1 (Remarks to the Author):

This work shows potentially biologic functional roles of ELAVL1a in zebrafish. Authors found that ELAVL1a is a LTA- and LPS-binding protein, capable of inhibiting the growth of various Gram-positive and -negative bacteria and explored the possible mechanism underlying the disruption of bacteria, including membrane depolarization and membrane permeabilization. Then they found that C-terminal 62 residues positioned at 181-242 of ELAVL1a is critical for the antibacterial activity. In addition, authors tested the antibacterial activity of ELAVL1a *in vitro* and *in vivo* in an attempt to try to confirm the results. However, many results in this work need to be validated, and the issues the authors should attend to as listed below.

Major concerns

1. Since the antibody used in the *in vivo* blockade assay was a mouse anti-human ELAVL1a antibody, it is important for authors to validate if this antibody works correctly in zebrafish, including the specificity and efficiency of this antibody for a zebrafish ELAVL1a homolog. Western blot assay is recommended for this procedure, in which the recombinant zebrafish ELAVL1a protein (rELAVL1a) and particularly the natural ELAVL1a proteins from zebrafish tissues and embryos should be included.

Response: Many thanks. We have added the Western blotting experiments to show the characterization of the antibody. As shown in supplemental Fig. 1, the Western blotting results showed that the antibody could not only react with the recombinant zebrafish ELAVL1a protein, the natural ELAVL1a proteins from zebrafish tissues (skin and muscle) and embryos, but also react with the natural ELAVL1a proteins from mouse tissue (muscle). These results were consistent with the description in the instructions, which claimed that this antibody could react with human, mouse and zebrafish ELAVL1a. Thus, the antibody was qualified for all the experiments in this paper. For the details, see the text and Supplemental Fig. 1, please.

2. The binding activity of zebrafish ELAVL1a protein to LTA, LPS and lipid A is the main discovery of the study. The authors claim that both LTA and LPS were recognized by ELAVL1a through a common C-terminal 62 residues positioned at 181-242, but in view of the large differences in composition and structure between LTA and LPS/lipid A molecules, it looks likely that nonspecific interactions may exist between ELAVL1a and LTA or LPS/lipid A. More solid evidence is needed to further exclude the possible nonspecific interactions and support the author's conclusion. It is recommended that various immunoprecipitation assays in different combinations should be added, including immunoprecipitation between the wild type ELAVL1a protein and LTA/LPS/lipid A, as well as between the mutant ELAVL1a proteins (deletion with C-terminal 62 residues and N/C-terminal RRM domains) and LTA/LPS/lipid A. Competitive binding assays should be included in the immunoprecipitation by introducing mutant ELAVL1a proteins, such as introducing the mutant ELAVL1a protein with deletion of C-terminal 62 residues in the experiment between the wild type ELAVL1a protein and LTA/LPS/lipid A. In addition, clarification of the detailed functional residues in the C-terminal 62-aa segment that contribute to the interaction of ELAVL1a with LTA/LPS/lipid A by charged and/or

hydrophobic interactions will be benefit to support the conclusion.

Response: Many thanks for the suggestion. As the routine (antigen-antibody) immunoprecipitation method is not suitable for the assay of the binding of ELAVL1a protein to LTA or LPS, we thus apply the pull-down experiment and competition binding assay to conform the specificity of the interaction of ELAVL1a protein with LTA and LPS. We have performed the pull-down assay which showed that rELAVL1a bound to both LTA- and LPS-conjugated resin column, but recombinant TRX-His tag did not. Furthermore, the competition assay showed that the interaction of rELAVL1a with LPS was clearly inhibited by either LTA or LPS in a dose-dependent manner. In addition, the competitive binding assays by introducing the truncated peptide E₁₋₁₈₀ (Which showed no affinity to LTA or LPS! See text please) in the interaction between the ELAVL1a or E₁₈₁₋₂₄₂ proteins and LTA or LPS showed that rELAVL1a and C-terminal 62 residues displayed strong affinity to LTA or LPS, and this was not inhibited by the truncated peptide E₁₋₁₈₀. These indicated the interactions between ELAVL1a and LTA or LPS/lipid A was specific. For the details, see the text and Supplementary Fig. 10, please.

For the second question, we have performed the site-directed mutation of ELAVL1a and bioactivity assay. The plasmid *pET28a/e181-242* was used for mutational analyses. Mutants were generated using Mut Express II Fast Mutagenesis Kit V2 (Vazyme, Nanjing, China) according to the kit's protocol. The mutated recombinant proteins were expressed and purified, and the antimicrobial activity and ligand binding assays of the mutant recombinant proteins were performed as described in text. The results showed that, compared with rE₁₈₁₋₂₄₂, the double mutation V192G/I193S (m3; hydrophobic Val and Ile to hydrophilic Gly and Ser) and the double mutation R203G/R204G (m4; two consecutive positively charged Arg to neutral Gly) resulted in a significant decrease in their antimicrobial activity against the bacteria tested (Fig. 8c and supplementary Fig. 18) as well as their affinity to LTA and LPS (Fig. 8d). These indicated that the hydrophobic residues Val192/Ile193 and the positively charged residue Arg203/Arg204 were at least part of the functional determinants contributing to the antimicrobial activity of rELAVL1a and the interaction of ELAVL1a with LTA and LPS. For the details, see the text and new Fig. 8 and Supplementary Fig. 18, please.

3. For bactericidal examination, plaque inhibition assay should be included in the study in addition to the growth inhibition test.

Response: Thanks for the suggestion. We have added another bactericidal examination, the Oxford cup agar diffusion method. For the Oxford cup agar diffusion, aliquots of 100 µl of 20 mM Tris-HCl buffer (pH7.4) containing 25, 35 and 50 µg/ml rELAVL1a were each added to an Oxford cup and then placed on agar plates containing 10⁵ cells of tested bacteria. The blank and negative controls were processed similarly, except that the rELAVL1a was replaced by 20 mM Tris-HCl buffer or 20 mM Tris-HCl buffer containing 50 µg/ml recombinant TRX-His tag peptide. As shown in Fig. 4b and Supplementary Fig. 12b, the rELAVL1a showed different size of halos in a dose-dependent manner in the agar diffusion test, while the rTRX-His tag peptide and Tris-HCl buffer did not. For the detail, see the text and new Fig.4 and Supplementary Fig. 12, please.

4. Subcellular localization analysis showed that the ELAVL1a protein distributes in the nucleus and cytoplasm, thus an intracellular bacterial infection model should be used instead of extracellular bacterial infection.

Response: Thanks for the suggestion. We have performed the experiment suggested, and used the *Edwardsiella tarda* (EIB202) as the intracellular bacterial infection model to test if the ELAVL1a had antibacterial activities against intracellular bacterium. As shown in new Fig. 4a and 4b, the growth of *E. tarda* was similarly inhibited by rELAVL1a in a dose-dependent manner. For the detail, see the text and new Fig.4, please.

5. In figure 2A, B, essential isotype antibody control data is missing. Please included these data in a supplementary figure, which is critical to exclude the non-specific binding of antibody to the zebrafish cells.

Response: Thanks for the suggestion. We have added the control data in the text and the supplementary, using the Mouse IgG1 Isotype Control antibody (ABIN457406) as control. As shown in Supplementary Fig. 2, no positive signal was seen in the control group using the isotype antibody as control instead of the primary antibody. These showed the binding of antibody to the zebrafish cells was specific. For the detail, see the text and supplementary Fig. 2.

6. In figure 5E, F the image is blurry. Please show high-quality merged images for the bacteria with FITC-labeled ELAVL1a. In addition, in figure 5G-L, results for the binding of TRX-His peptide to LTA, LPS.....are missing.

Response: Thanks for the suggestion. For the first question, we have repeated the experiment and the images were zoomed in to make sure that the bacteria were clearly visible, and we also have given the merged images. In addition, as suggested by another reviewer (see below), we have moved the data to the supplementary. For the details, see the new supplementary Fig. 9, please.

For the second question, we have repeated the experiments, and added the TRX-His peptide as another control. The results showed that the rELAVL1a bound to both LTA and LPS in a dose-dependent manner, whereas BSA or TRX-His peptide used as negative control did not. For the details, see the text and new Fig. 3, please.

7. Again, in figure 6A, essential control data is missing. Since the ELAVL1a is fused with TRX-His tag, it is critical to add the control experiments to exclude the effect of TRX-His tag. Or you can cut the TRX-His tag with corresponding enzyme.

Response: Thanks for the suggestion. We have repeated the experiment and added another control group, using the TRX-His tag instead of rELAVL1a, to exclude the effect of TRX-His tag. The results showed that the TRX-His tag had no antibacterial activities against the bacteria tested. For the detail, see the new Fig 4, please.

8. In figure 8C, authors should add the rescue experiment to confirm the specificity of MO.

Response: Thanks for the suggestion. We have added the rescue experiment to confirm the specificity of *elav11a*-MO. For rescue experiments, the open reading frame (ORF) of *elav11a* was cloned into vector pCS2, and ORF mRNA was synthesized by mMACHINE mMESSAGE SP6 Kit (Life Technologies, Carlsbad, CA). Acquired ORF mRNA was co-injected with the relevant *elav11a*-MO. The results showed that the synthesis of ELAVL1a was reduced in the embryos microinjected with *elav11a*-MO, indicating a successful and specific knockdown of ELAVL1a (Fig. 6cA). As shown in Fig. 6cB, the 24-h mortality of *elav11a*-MO-knockdown embryos increased up to 79%, in contrast to 63% mortality of control embryos, after challenge with live *A. hydrophila*,

and this increase in mortality could be rescued by co-injection of *elavl1a*-MO with *elavl1a*-mRNA. For the detail, see the text and new Fig. 6, please.

Minor concern

9. In figure 7A, what does “control” specifically mean?

Response: Sorry for the missing of the indication of “control group” in Fig 7a. The HEPES buffer containing 20 mM glucose was used as control in this experiment, and we have added this information in the new figure legend. For the detail, see the new Fig 5, please.

10. English writing needs to be improved.

Response: Thanks for the suggestion. We have read the text carefully and improved the English writing.

11. Authors described ELAVL1a as a pattern recognition receptor in abstract and discussion sections, there is not enough evidence for this conclusion.

Response: Many thanks. We have provided more evidence for ELAVL1a acting as a pattern recognition receptor in the discussion section. For the detail, see the text, please.

Reviewer #2 (Remarks to the Author):

Brief summary:

The current study talks about identification and characterization of zebrafish ELAVL1a which is an analog of mammalian HuR. ELAVL/Hu are a family of RNA binding proteins which regulate gene expression at post transcriptional level and are necessary for proper embryonic development. With a robust analysis of experiments, they go on to demonstrate that a function of ELAVL1A is to provide immunity from bacteria during embryonic development. The authors state and prove that ELAVL1a binds to LPS and LTA specifically on Gram-negative and Gram-positive bacteria and disrupts the membrane via depolarization and permeabilization. They also demonstrate that the C-terminal 62 residues from position 181-242, which does not contain any RRM, were vital in providing this anti-bacterial activity. Furthermore, inhibition of ELAVL1a via antibody or gene knockdown resulted in increased mortality as the pathogen resistant activity was reduced. Collectively, this work shows that ELAVL1 is an immune factor in zebrafish embryos that can protect from bacterial infection.

Overall impression of the work:

The authors of this paper have done a fine job of combining computational sequencing and modelling with protein purification and came up with a robust body of work which can influence others in its stead. The quality of data is adequate in most cases accompanied with appropriate controls. There is very little room for criticism on the amount of work done and no need for additional experiments to be done to corroborate their findings. However, there is a significant amount of data that should be moved to the supplement or completely removed from the paper. Also, certain data need to be quantified in order to be presented and the statistical tests used must be clearly named in the legend so that the reader may determine their appropriateness. There are also several improvements that can be made to the way the article is organized and formatted. The authors should carefully proofread their

work and change some figures to make it easier for the reader to follow. The following comments on the manuscript should help to guide them in improving their study.

This work will provide more context to the previously undefined role of HuR family of proteins in providing immunity during embryonic development.

Major Concerns

1. Figure 1.

(1) It is concerning that the sequence coverage for the protein ID of band 5 was only 7% of the ELAVL1a sequence. This may be because it seemed to be one of the minor bands. Was there a control for non-specific binding done in this experiment? It would be best to show pull down result using un-conjugated Sepharose as a control.

Response: Thanks. We have added the entire gels in Figure 1a, and the lane 6 of this new figure showed the result using un-conjugated Sepharose as a control. For the detail, see the new Figure 1, please.

(2) There is a lot of wasted space in this figure. I would suggest rearranging by placing the sequence coverage information in the text or legend and including to Fig 3a and 3b panels in this figure. The flow should be as such: 1a. pull-down gel, 1b. ELAV1a domain structure, 1c. ELAV1a 3D structure prediction, 1d. organ blots, 1e. stage blots. The phylogenetic tree 3c. can be moved to the supplement.

Response: Many thanks. We have placed the sequence coverage information in the text and rearranged the Fig. 1 as suggested. The re-arranged the Fig. 1 included: 1a. the new pull-down gel; 1b. ELAV1a domain structure; 1c. ELAV1a 3D structure prediction; 1d. organ blots; 1e. stage blots. In addition, we have moved the phylogenetic tree to the supplementary. For the detail, see the new Fig. 1 and Supplementary Fig. 6.

(3) Actin bands look not consistent. Especially in b, it looks underexposed. The quantification of the Western blot can just be numbers below the band. That way it is easier to interpret it rather than in graphical format.

Response: Many thanks. We have repeated this experiment to make sure that all the actin bands look consistent. We have put the number of the quantification of the Western blot below the bands. For the detail, see the new Fig.1, please.

(4) Strike the statement in line 74-75 that ELAVL1a is relatively higher in the ovary. The bands for heart, ovary and testes are all oversaturated so differences in abundance in these organs is impossible to determine from these data.

Response: Many thanks. We have modified the description “with relatively higher level in the heart, ovary and testis (Fig. 1d)”, and removed “This suggested that ELAVL1a was abundantly distributed in the ovary”. For the detail, see the text, please.

2. Figure 2.

(1) 2a: the green signal in the 488 channel is completely oversaturated

Response: Many thanks. We have repeated the immunohistochemical analysis to make sure that all the channel look better. For the detail, see the new Fig. 2, please.

(2) Figure 2b, the green and blue (Alexa 488 and DAPI) fluorescence looks inconsistent between A, B and C as if taken at different exposures.

Response: Many thanks. We have removed the panels Fig 2b and 2c as suggested below, and

modified the related descriptions in the “Results” section and “Methods” section. For the detail, see the text and new Fig. 2, please.

(3) Fig 2b and 2c: without quantification this data is not convincing. I do not see strong evidence of nuclear localization in these figures. These two panels are unnecessary for the main conclusions of the paper so I would suggest removing them entirely unless quantification data is available.

Response: Many thanks. We have removed the panels Fig 2b and 2c, and modified the related descriptions in the “Results” section and “Methods” section. For the detail, see the text and new Fig. 2, please.

(4) After removing the localization data in 2b and 2c, I would suggest combining 2a with figure 4 to make a complete figure 2 about the expression of ELAV1a in different stages and tissues.

Response: Many thanks. After removing the panels of Fig 2b and 2c, we have combined the Fig 2a with Fig 4. For the detail, see the new Fig. 2, please.

3. Figure 3.

Combine into figure 1 as suggested above

Response: Many thanks. We have combined the Fig 3 into Fig 1. For the detail, see the new Fig 1, please.

4. Figure 4.

(1) Combine into figure 2 as suggested above

Response: Many thanks. We have combined the Fig 4 into new Fig 2. For the detail, see the new Fig 2, please.

(2) For y axis legends change to Relative ELAV1a expression

Response: Many thanks. We have changed the y axis legends to “Relative *elav1a* expression”. For the detail, see the new Fig 2, please.

(3) Figure 4c: WISH color seems diffused and not specific. Is it the blue that we’re supposed to focus on or the purple? Not clear.

Response: Sorry for the confusion we have made. The color we should focus on is purple, and we have added the description of “The color of the positive signal is purple” in the legend of new Fig 2. For the detail, see the new figure legend of new Fig 2, please.

5. Figure 5 (Now Figure 3)

(1) Figure 5e and f: The fluorescence and the scale bars do not seem very clear. You can barely see the bacteria. It would be better if the images were zoomed in and the bacteria were clearly visible (100X magnification). Also, these data should be quantified to be impactful. If there is no quantification, this data needs to be moved to the supplement.

Response: Many thanks. For the first question, we have repeated the experiment and the bacteria were observed and photographed under the Leica confocal microscope to make sure that the bacteria were clearly visible. For the second question, we have moved the data to the supplementary. For the detail, see the new supplementary Fig. 9, please.

6. Figure 7 (Now Figure 5)

(1) Figure 7a: The experimental set up was not clearly explained in the main text or legend. What is the fluorescence intensity being measured? How does it demonstrate loss of membrane polarization? Please provide references.

Response: Thanks for the suggestions. For the first question, the changes in fluorescence intensity were recorded with a Tecan GENios plus spectrofluorometer at an excitation wavelength of 622 nm and an emission wavelength of 670 nm. We have added the information in the figure legend of new Fig. 5.

For the second question, after interaction with intact cytoplasmic membrane, the fluorescent probe DiSC₃-5 was quenched. After incubation with the antimicrobial protein or peptide, the membrane depolarization was induced the probe was released to the medium, ensuing in an increase of fluorescence that can be quantified and monitored as a function of time. We have added relative reference in the text. For the detail, see the text, please.

(2) Indicate the statistical test used to determine p values in the legend.

Response: Thanks for the suggestion. We have added “The significance of difference was determined by one-way ANOVA.” in the figure legend of Fig. 5.

7. Figure 8 (Now Figure 6)

(1) Indicate the statistical test used in the figure legend and show error in both directions

Response: Many thanks. We have added “The significance of difference was determined by one-way ANOVA.” in the figure legend of Fig. 6, and shown the error bars in both directions. For the detail, see the new Fig legends and the new Fig 6, please.

(2) What was the MOI of the challenge with *A. hydrophilia*? This needs to be indicated somewhere in the legend. It appears to be too high as still over 50% of embryos die even with injection of exogenous ELAVL1a.

Response: Many thanks. For the first question, the density of *A. hydrophilia* used to microinject to embryos was 8.3×10^7 cells / ml, and we have added this information in the figure legend of new Fig 6.

For the second question, the 24-h mortality rate of the embryos injected with rELAVL1a was reduced to 54.4%, which was markedly lower (statistically significant) than that of the embryos injected with PBS, BSA and AcAb alone (68.9%, 69.2% and 69.3%). Although it was still over 50%, this might be due to the low amount of proteins injected.

(3) The experiment in 8d. is an odd way to show that the bacteria were being killed. This is normally done by CFU assay as bacterial DNA can be present even when no viable bacteria are present. There is also no loading control. This data should be moved to the supplement or removed entirely.

Response: Thanks for the suggestion. We have removed the data. For the detail, see the text, please.

8. Figure 9 (Now Figure 7).

(1) Panel B should be moved to the supplement.

Response: Thanks. We have moved the panel B of Fig 9 to the supplementary. For the detail, see the new Fig. 7 and new Supplementary Fig.15, please.

(2) Please indicate the statistical test used in the legend.

We have added “The significance of difference was determined by one-way ANOVA.” in the

figure legend of Fig. 7.

Minor concerns:

1. For all graphs error bars need to be shown in both directions.

Response: Thanks for the suggestion. We have modified all figures to make sure that all graphs error bars were shown in both directions.

2. Line 23: should be biological instead of biologic

Response: Thanks. We have corrected this writing error.

3. Line 228: Here instead of Her.

Response: Thanks. We have corrected this writing error.

4. Line 296: verify instead of verity.

Response: Thanks. We have corrected this writing error.

5. Figure 2a and b: Alexa 488 should be renamed to ELAVL1a as that's what we're observing rather than the channel.

Response: Thanks. We have changed the "Alexa 488" to "ELAVL1a" in Fig 2. For the detail, please see the new Fig 2.

6. Figure 5a and d: These can be moved to supplement.

Response: Thanks. We have moved the Fig 5a and 5d to the supplementary and rearranged the figure which now is Fig 3. For the detail, please see the new Fig 3.

7. Figure 5a, b, c and d: Labelling them appropriately in the figure rather than in figure legend would be clearer. Label blots above with +/- for the added ELAV1a and the bacteria input instead of 1-7.

Response: Thanks for the suggestion. We have redrawn the figure which now is Fig 3, and the labels were placed in the figure by the way suggested. For the detail, see the new Fig 3, please.

8. Line 569: Gram-positive instead of Gran-positive.

Response: Thanks. We have corrected this writing error.

9. Figure 6a: Some graphs could be moved to supplement. Just one bacterium from Gram-positive side and one from Gram-negative side would be enough.

Response: Thanks for the suggestion. We have moved some graphs to the supplementary, and only the Gram-positive *Micrococcus luteus* and Gram-negative *E. coli* were remained. For the detail, see the new Fig 4, please.

10. Figure 7 caption: membranolytic is not a word.

Response: Thanks. We have changed it to "membrane lytic". For the detail, see the new figure legends of Fig. 5.

11. Figure 7a and b: Same as comment 9.

Response: Thanks for the suggestion. We have moved some graphs to the supplementary, and only the Gram-positive *Micrococcus luteus* and Gram-negative *E. coli* were remained. For the detail, see the new Fig 5, please.

12. Figure 8a: Same as comment 9.

Response: Thanks for the suggestion. We have moved some graphs to the supplementary, and only the Gram-positive *Micrococcus luteus* and Gram-negative *E. coli* were remained. For the detail, see the new Fig 6, please.

13. Figure 9c: Same as comment 9.

Response: Thanks for the suggestion. We have moved some graphs to the supplementary, and only the Gram-positive *Micrococcus luteus* and Gram-negative *E. coli* were remained. For the detail, see the new Fig 7, please.

14. Known functions of mammalian HuR protein are quite different from ELAVL1a as listed in this paper. It would be helpful to know if Zebrafish ELAVL1a has similar functions to its mammalian counterpart

Response: Thanks. Yes, Known functions of mammalian HuR protein are quite different from ELAVL1a. Unfortunately, information regarding ELAV1/HuR/HuA function is currently not available in other animal models including zebrafish, thus we do not know if Zebrafish ELAVL1a has similar functions to its mammalian counterpart. This deserves further study in the future.

REVIEWERS' COMMENTS:

Reviewer #1 (Remarks to the Author):

All the questions had been addressed, it suggests to be accepted.

Reviewer #2 (Remarks to the Author):

Thank you for your careful revision of this manuscript. The authors have adequately addressed my concerns and I think the manuscript is significantly improved. I have only one remaining concern that need to be addressed:

In line 148-149 the authors state that "These (data?) indicated that ELAVL1a was (a) bactericidal agent capable of directly killing Gram-positive and Gram-negative bacteria." Because the authors did not perform a bactericidal assay, it cannot be definitively concluded that ELAVL1a is bactericidal. The Oxford cup diffusion method is able to test the ability of a compound to inhibit bacterial growth and is used for determining minimum inhibitory concentrations (MICs) but not minimum bactericidal concentrations (MBCs). In order to demonstrate conclusively that a compound is bactericidal, a colony forming unit assay must be performed where the bacteria are treated with the compound and then replated to see if any live bacteria remain. Alternatively a live/dead staining method coupled with flow cytometry can be used to measure bacterial killing. However, I do not think it is important for the major conclusions of this paper to distinguish between inhibitory and killing effects. The authors could address this concern by simply revising their statement in lines 148-149 to state that "ELAVL1a is an antibacterial agent capable of directly inhibiting growth of Gram-positive and -negative bacteria".

To Reviewer #1:

Remarks to the Author:

All the questions had been addressed, it suggests to be accepted.

Response: We thank the Reviewer for the positive feedback.

To Reviewer #2:

Remarks to the Author:

Thank you for your careful revision of this manuscript. The authors have adequately addressed my concerns and I think the manuscript is significantly improved. I have only one remaining concern that need to be addressed:

In line 148-149 the authors state that "These (data?) indicated that ELAVL1a was (a) bactericidal agent capable of directly killing Gram-positive and Gram-negative bacteria." Because the authors did not perform a bactericidal assay, it cannot be definitively concluded that ELAVL1a is bactericidal. The Oxford cup diffusion method is able to test the ability of a compound to inhibit bacterial growth and is used for determining minimum inhibitory concentrations (MICs) but not minimum bactericidal concentrations (MBCs). In order to demonstrate conclusively that a compound is bactericidal, a colony forming unit assay must be performed where the bacteria are treated with the compound and then replated to see if any live bacteria remain. Alternatively a live/dead staining method coupled with flow cytometry can be used to measure bacterial killing.

However, I do not think it is important for the major conclusions of this paper to distinguish between inhibitory and killing effects. The authors could address this concern by simply revising their statement in lines 148-149 to state that "ELAVL1a is an antibacterial agent capable of directly inhibiting growth of Gram-positive and -negative bacteria".

Response: Many thanks. We fully accept the proposal, and we have revised the descriptions in lines 148-149 to "ELAVL1a is an antibacterial agent capable of directly inhibiting growth of Gram-positive and -negative bacteria". For the details, see the text, please.